# Pedagogy Pro-Design and Climate Literacy: Teaching Methods and Research Approaches for Sustainable Architecture

**Rosa Schiano-Phan [1,\*], Joana C. S. Gonçalves [1,2] and Juan A. Vallejo [1]**

[1] School of Architecture and Cities, University of Westminster, London NW1 5LS, UK; j.soaresgoncalves1@westminster.ac.uk (J.C.S.G.); j.vallejo@westminster.ac.uk (J.A.V.)

[2] Faculty of Architecture and Urbanism, University of Sao Paulo, Sao Paulo 05508-080, Brazil

\* Correspondence: r.schianophan@westminster.ac.uk

**Abstract:** There is a genuine concern that the current level of sustainability education provided in the mainstream architectural curricula is no longer sufficient to combat urgent climate challenges, and that a stronger interdisciplinary approach needs to be followed where architectural students are formed and empowered with a different pedagogical paradigm, better tools, and diverse sets of skills. This paper examines the various pedagogical approaches to the teaching and learning of environmental design principles and practice in architectural education with a focus on recurrent methods applied in specialist curricula in the UK. An in-depth analysis of a pedagogical case study based on the eight-year experience of the Architecture and Environmental Design MSc course at the University of Westminster is presented with reference to the pedagogical methods identified in the literature. A reflective exercise based on the specific methods adopted in the course and examples of students' outputs and experiences allows a critical evaluation of the pedagogical case study. The paper concludes by highlighting the challenges and opportunities of introducing climate literacy at postgraduate level and the benefits of adopting a research-led approach based on collaborations with industry partners.

**Keywords:** environmental design; sustainable architecture; pedagogy; climate literacy; environmental architecture; curriculum; research





## 1. Introduction

The latest strategic international goals identified by Advance HE for 2021–2024 [1], align with the UN Sustainable Development Goals and highlight the higher education (HE) sector's commitment to sustainable development and climate action. Research-informed teaching and actively engaging students in research have always been a prime concern within HE [2]. This becomes particularly relevant in the field of architecture where shortfalls in the HE curriculum have been extensively reported [3,4]. Courses in environmental design and evidence-based architectural design have become part of the HE offerings in many UK schools of architecture over the last few decades. As a vehicle to teaching and learning (T&L) of environmental design, the use and effectiveness of the evidence-based design approach [5], the integrated studio [6,7], and the live projects [8] have been documented and discussed by several scholars over the last 20 years.

In this article, the authors present a methodological approach to environmental design education examined and applied through the case study of an intensive postgraduate (MSc) course. The proposed learning method, based on the substantial integration of performance assessment methods throughout the design process, aims at forming professionals, encompassing interdisciplinary skills across architecture and engineering, who are empowered with technical skills and inherently invested with agency of change.

The article will examine this transformative process which takes place through didactic projects and direct engagement with applied research and industry. Reference to the various

stages of the didactic process and 'learning-by-doing' approach, such as evaluation studies, design projects, and dissertation projects will be made and evaluated in the context of their outcomes, impacts, and applied feedback protocols. This article will also highlight the crucial role that the collaborative research with industry partners has on the students' rapid development of professional skills and advanced technical knowledge, fit for the current and future expectations of the real world of professional practice. In addition to that, emerging concerns related to air-pollution and well-being in the built environment have led to innovative research topics contributing to knowledge ranging from the development of analytic tools, monitoring devices, and construction systems to the design of zero-carbon buildings and sustainable master planning examples. Some of these projects have been implemented in real practices, validating the success of this learning method and enhancing the students' learning experience.

## 2. Objectives

The article aims to present a pedagogical case study for the integration of climate literacy in specialist architectural education. The objective is to share the methodological approach of a postgraduate course focused on the relationship between architecture and environmental design in the context of the existing recognized teaching and learning methods pertinent to the subject matter and to reflect on the challenges and opportunities of introducing climate literacy on a postgraduate level.

## 3. Methodology

This article is based on pedagogical research developed over the course of eight years at the School of Architecture and Cities, University of Westminster. The research was based on the collection of primary and secondary data and observations related to the design, development and implementation of a new MSc curriculum in Architecture and Environmental Design (AED). The methodology adopted for this research is based on three main phases:

1. Review of the theoretical background and relevant pedagogical references and creation of an evaluation framework (see Section 4).
2. Presentation and analysis of a pedagogical case study based on the experience of the MSc AED (the case study) and identification of innovative and experimental features (see Section 5).
3. Reflection and critical appraisal of the challenges and opportunities offered by the proposed pedagogical model and applicability to wider contexts (see Section 6).

The primary research method is qualitative analysis based on the case study method [9,10]. The limitation of this method is clearly the focus on one case study only, which could present some biases and peculiarity, potentially reducing the wider applicability of the results and outcomes. However, given the strong links that the presented teaching and learning methods have with traditional architectural education, it is expected that the level of applicability of the case study learnings could be quite high among UK and international HE institutions.

## 4. Theoretical Background

Embedding research in universities' curricula and facilitating students' first approach to research has always been high on the agenda of HE institutions [2]. As a vehicle to teaching and learning of architecture in general and environmental design more specifically, the use of the integrated studio, live projects, and the evidence-based design approach have been often discussed among scholars over the last 20 years. These practices are gaining significant popularity across HE institutions and their potential has been discussed in an increasing number of publications [8,11–13].

Since the IXX century, and the introduction of the Beaux-Art's atelier system, the main vehicle for the delivery of architectural education has been the design studio [14]. To these days, the design studio is synonym of blended and project-based learning [15],

where the students apply theory into practice in a process also referred to as 'learning by doing' [16]. This ties in with the concept of integrated studio [7], which highlights the importance of finding a strong connection between the taught modules, where the principles of building technology and environmental design are introduced and where the numerical and computation skills are transferred, and the design studio modules, where these principles and skills are applied and practiced. This process is quite challenging and, as Szokoloy [7] points out, it is rarely fully integrated due to disconnections between the various learning environments and teaching staff often differing between the studio and the technical studies in terms of skills and pedagogical objectives.

Live projects embed the principles of contemporary experiential learning theories [17,18], providing students with real-life experiences and opportunities to learn through direct engagement with activities relevant to their subject of study. Live projects are also a very useful tool to involve students and staff with external communities, and to build new audiences with clear benefit to the course outreach. Live projects engage students in active learning [19], developing subject-specific employability skills and improving the level of student participation and outcomes. Live project pedagogies [20–22] are widely applied by several Schools of Architecture and associated environmental design courses in the UK (e.g., The University of Nottingham, the Architectural Association [23], Cardiff Welsh School of Architecture [24], and the University of West of England [6] to name a few).

The evidence-based design approach to architectural education [5] is intrinsic to the ethos of curricula which aims to develop climate literacy and a greater level of numeracy in support of the collection of evidence and of its interpretation through empirical and analytical methods. Within such approach, students are encouraged to question every single design decision based on data gathering and to justify the creative design process with a logical and demonstrable argument underpinned by climatic, environmental, and social investigations. Based on the theoretical framework identified above, the pedagogical case study exemplifies the practical application of these concepts and the critical analysis of challenges and opportunities related to the delivery of a postgraduate course in environmental design.

## 5. The Pedagogical Case Study

The Architecture and Environmental Design (MSc) course was specifically designed to empower architects and building industry professionals with the knowledge and skills to design built environments, which are climatically responsive, and which have minimal energy demand by maximizing reliance on passive heating, cooling, ventilation, and daylighting strategies, thus improving thermal, visual, and acoustic comfort [25–27], as well as occupants' health and wellbeing [28]. The approach of the course was, from the start, aimed at adopting an evidence-based design methodology overlaid on the standard design methodology centered around a brief traditionally inspired by a primary socio-cultural agenda. The conventional design methodology, generally taught in most western schools of architecture, starts from a qualitative analysis of context, site, and users, leading to the identification of generative design guidelines producing rationales for building massing and subsequent design development (from schematic design to detail design, depending on the extent of the brief) [29,30].

The evidence-based environmental design and climate-responsive approach is in fact much more ambitious than a simple overlay. The basis of the methodology, and the focus of the brief in the ED project, is expanded to include both the socio-cultural agenda and the environmental agenda. The starting point for this revisited methodology is no longer solely a qualitative site and context analysis but a detailed climate analysis, which then stems into the site and microclimate analysis and, at that point, searches for a combination of human/socio-centered and climatic/micro-climatic generative guidelines for the so-called bioclimatic design [31]. Underpinning this process are the students' numeracy skills which enable the justification of the various design moves with adequate data analysis, collected from onsite fieldwork and generated through computational modelling and simulation.

In fact, one of the aims of the course is to effectively retrain architects and other building professionals to adopt a new methodological focus as well as the high numeracy and analytical skills, required to not only perform fieldwork and simulation analyses but most importantly to convert them into communicable design decisions. In this context, it is worth mentioning that the skill sets of the lecturers and tutors are also important, and they must themselves be trained and specialized in the environmental design discipline with an architectural sensitivity. So far, the course employs staff from both the architectural and engineering backgrounds. The size of the course is also important and, like many master courses in the UK, which go above and beyond the architectural professional accreditation, this course has a target size of about 20 students per year, which of course can increase, making sure to have a high ratio of tutors per students, in order to maintain close one-to-one tutoring opportunities and teaching and learning quality.

Although, over the last few decades, efforts are being made to embed a more environmentally sensitive and climate conscious approach to design from the early stages of undergraduate architectural education in many UK HE institutions, there are still many architects around the world who have little or no training at all in environmental design quantitative methods [3]. Therefore, there is a great deal of potential in retraining those professionals with these skills and basic understanding of climate-responsive design. The traditional architecture T&L methods based on design studio workshops, reference to precedents study, frequent formative critical feedback (through weekly tutorials and periodic 'crits'), and coursework based on portfolios of drawings are supplemented with additional and often engineering-based T&L methods, such as classroom-based lectures, environmental laboratory experiments, intensive performance analysis software workshops, and coursework based on technical report writing, which create a stimulating, albeit demanding, interdisciplinary learning environment.

The course combines taught modules and design studio modules (Figure 1) delivered by semesters with a thematic continuity, which sees topic-centered weeks when core principles and subjects are introduced in the taught modules and followed up in the software workshops and expected to be applied in the project development for the following week. This structure exemplifies the epistemological approaches, which can be considered primarily pragmatic, examining problems of discrete to holistic nature [5]. At the same time, this structure stimulates cognitive processes which go from the analytic to the synthetic, typical of the left and right side of the brain [32].

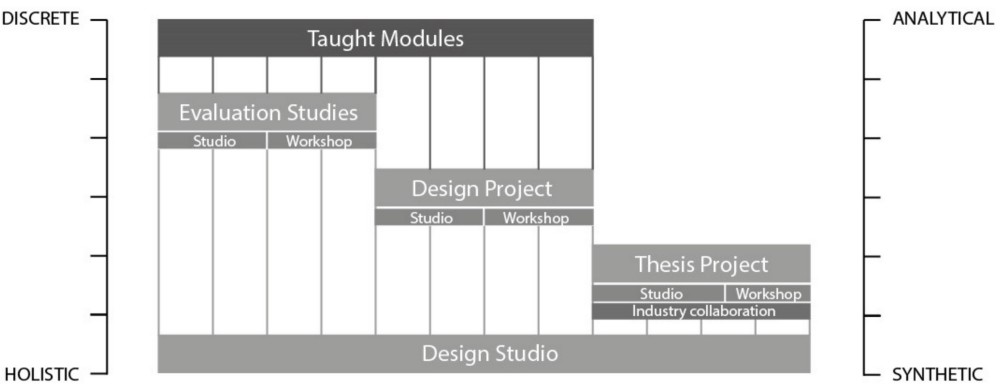

**Figure 1.** Diagram explaining the sequence of learning methods used in the course against epistemological approaches (discrete to holistic) and cognitive processes (analytical to synthetic).

Apart from taught modules in semester one and two, over 60% of the course credits are achieved through design studio-based projects. The design studio projects are: Evaluation Studies (semester one), Design Project (semester two), and Thesis Project (semester three). The rationale behind the succession of these design studio modules is in the gradual introduction of principles through the observation and evaluation of building performance of real buildings (Evaluation), which is undertaken by students in groupwork, followed by

the application of principles in the design process (Design Project), which is undertaken partially in group and partially individually, and finally the application of all the acquired knowledge to conduct design-based research and demonstrate complete independence in critical and methodological design skills (Thesis Project). The Thesis Project stage in the MSc AED is also that of engaging with the industry and finding common research agendas with partnering engineering and architectural practices. A detailed overview of these crucial phases of the pedagogical approach offered by the MSc AED course, including examples of students' outputs, is outlined below followed by a discussion of the way that they implement the theoretical framework of the live project, integrated studio, and evidence-based design approaches.

It is worth clarifying that the examples of course work included in the paper for semesters one and two—i.e., Evaluation Studies and Design Project—are not from the same year; for this reason, the two pieces of students' work are not linked, as it normally happens in the flow of a specific academic year. This was carried out to give an idea of the variety of projects that have developed in the course, including projects in sites outside London (where the course is located), in different climates—in this case, the city of Sao Paulo, in Brazil.

### 5.1. Evaluation Studies

As the first semester project of the course, the Evaluation Studies project is dedicated to the analysis of the environmental performance of existing built environments, encompassing buildings, and outdoor spaces, in different urban microclimates, looking at specific architectural features, building functions, and occupants' behavior patterns, by means of fieldwork and analytic studies. This empirical and analytic research-based project explores the mutual effects that the above-mentioned factors have on indoor and outdoor environmental quality, human comfort, and buildings' energy demand. Considerations on other factors such as health and well-being and the impact of materials choices on embodied carbon of buildings and outdoor spaces can also be included in the evaluation process.

### 5.1.1. Didactic Aims and Methods

The key learning objective and overarching scope of the Evaluation Studies are: (1) to provide students with the skills and methods to evaluate the performance of existing buildings, and space in between buildings, and (2) to provide a platform for learning from observation and technical analysis of existing indoor and outdoor environments. The aim is to draw lessons from real case studies, which will inform design approaches and strategies developed in the subsequent Design Projects. Typically, the case study areas and buildings are chosen in London, where the course is located. However, during the pandemic and the consequent on-line teaching, students were asked to identify case studies from their own home locations (academic year 2020–2021), which involved collecting examples from a variety of climates and geographical locations.

The methodological approach entails fieldwork and analytical studies, starting from the analysis of the climate, followed by that of site and microclimate. At this stage, fieldwork involves spot and continuous measurements (of a recommended period of 2 to 3 weeks minimum) of key environmental variables (air temperature, humidity, air movement, solar access, sky illuminance, air and noise pollution). Analytically, the studies include the assessment of the effects of urban morphology, building form, and orientation on solar and daylight access versus potential risks of the overshadowing and, or overheating, air movement around buildings and the overall conditions of outdoor comfort. All these studies combined allow the initial understanding of the potential positive and negative impacts of climatic and microclimatic conditions upon outdoor as well as indoor environments.

For the assessment of the case study building and its internal spaces, fieldwork consists of measurements in situ of the same environmental variables measured outdoors. Occupancy and occupants' activity are to be mapped over time and in different parts of the site, alongside the records of environmental variables. Ideally, the measurements are

accompanied by administering a post-occupancy evaluation survey and interviews with inhabitants and, where appropriate, with designers and site managers. Fieldwork including measurements in situ and occupants' survey are aligned with the guidelines established by specialized references on post-occupancy evaluation [33–36].

Once indoors, the observation of how and when spaces are occupied, and by whom, as well as the spatial and temporal distribution of equipment usage are key for the process of mapping the influence and the role of occupants upon the overall buildings' environmental performance. Furthermore, the fieldwork should address the following questions: How do occupants define desirable or acceptable environmental conditions for themselves? How does the site serve the requirements of its inhabitants and wider public? How do occupants vary environmental conditions for themselves? How feasible or desirable is it for them to be able to influence environmental conditions? In addition to that, the scenario of the pandemic imposes extra challenges for the occupants' overall comfort and environmental quality, giving the longer permanence indoors, the previously unplanned multipurpose use of certain spaces (exclusive living environments becoming work and study spaces as well), and the need for fresh-air intake given the importance of air quality, among other questions and uncertainties associated with the future of buildings [37–40]. In this respect, questions about these specific issues are encouraged to be raised, identifying the ways in which the occupants can cope with such challenges and adapt to their new environmental and spatial requirements.

At the end of the outdoor and indoor fieldwork, and the preliminary analytical work related to climate, microclimate, and site conditions, a new set of specific research questions are to be formulated based on environmental issues associated with both the building design and occupational patterns, creating the research agenda that will give focus to the analytical phase of the Evaluation Studies.

The indoor analytical studies focus on a critical evaluation of thermal and visual comfort throughout the year, addressing specific issues and questions identified during the fieldwork. In this respect, a series of parametric studies involving dynamic thermal simulations inform the relative importance of occupancy and architectural/physical parameters on the building's energy balance and consequent thermal response in specific days of the year, following scientific and technical rigor suggested by the specialized literature [41,42]. Daylight analytical assessment also considers yearly performance as well as specific illuminance conditions at particular days and times [43]. Simplistically, building parameters examined at this stage of the study include form and orientation, materiality, size, and distribution of internal and transitional spaces and specific features of the building's envelope, with special attention to the distribution of glazing areas, apertures, and presence or absence of shading devices, as suggested by Goncalves et al. [41].

The analytical assessment of the case study building encompasses manual calculations to determine sun altitudes, sky view and daylight factors, thermal comfort zones, and/or natural ventilation flow rates; simplified computational studies to determine incident solar radiation on surfaces, external wind flow, and/or indoor luminance and illuminance levels for a selected moment in time; and advanced analytical dynamic studies of thermal and daylight performance to allow an hourly evaluation of indoor thermal comfort, building energy balance, and/or spatial daylight autonomy based on the pre-established research agendas.

In the final stage of the Evaluation Study, results from fieldwork and analytical work are compared and the environmental performance of the case study (including internal and external spaces) is measured against benchmarks, when appropriate. To conclude, the main lessons from the case study and how it may inform future design research are to be identified and related to alternative design solutions that could improve the performance of buildings and open spaces in the current climatic context and in the future of climate change.

The Evaluation Study is structured to be developed in groups along a 12-week program, in which the first 5 weeks include climate analysis, fieldwork activities, and the formulation of the research agendas, followed by a 7-week period of analytical assessments and the elaboration of related recommendations for potential refurbishment strategies.

5.1.2. Students Outputs

Over the years, the briefs for the Evaluation Studies produced a diverse set of students' outputs, which typically consisted in a technical report illustrating the fieldwork and predictive performance analysis of the sites and buildings of interest for that year. The building typologies varied year on year with mostly residential buildings (Academic Year 2014/15 to 2018/19), office buildings (Academic Year 2019/20), and lately 'Work and live environments during the pandemic'. As part of the studio on the 'Environmental Performance of Modernist Buildings in London', run for three years, the students extensively analyzed a very successful example of modernist architecture: The Golden Lane Estate (Figure 2).

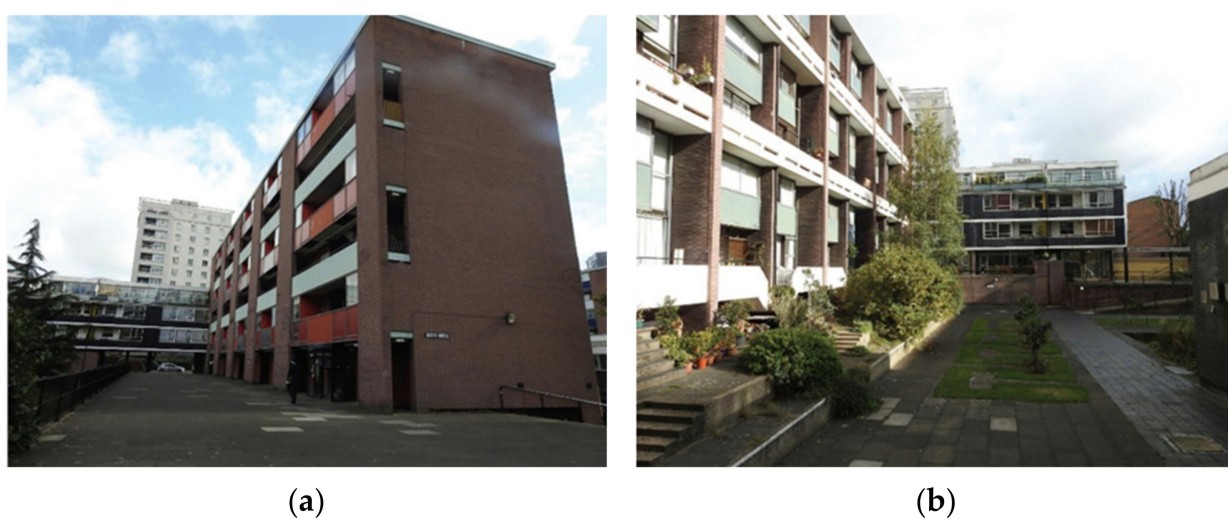

(**a**)                                                    (**b**)

**Figure 2.** Golden Lane Estate: (**a**) the Bayer House front façade; (**b**) the Bayer House south façade and back courtyard (source: Sharmeen Khan, student 2014–15).

The London housing development was an extremely useful learning vehicle due to the opportunity for site visits by the students and the engagement with residents. Over the years, the complex was so successfully studied that it led to several peer-reviewed publications co-authored by staff and students [44–46]. The Golden Lane Estate is a Grade-II-listed, high-density, low-cost housing complex built after the Second World War and designed by three young architects: Peter Chamberlin, Geoffrey Powell, and Christof Bon. It was built over a bombed site and embraced the post-war modern architecture ethos, environmental considerations, and inclusion of social facilities and landscaped communal spaces. The development survived 60 years mostly unchanged in its configuration, and adapted well to the demands and evolving conditions of a cosmopolitan capital such as London. The main difference, however, is in the typology of residents which reside in the estates. As a result of gentrification, these changed from social workers renting from the council to more affluent professionals, who can afford to rent or buy following the privatization of the 1980s. The students were tasked to analyze the environmental performance of the open spaces and selected apartments in the various buildings. The exercise allowed them to apply an interdisciplinary approach to the analysis, combining the awareness of the social and historical context of a very specific architectural period with the evidence-based approach, which allowed them to demonstrate the positive environmental attributes as well as performance shortfalls.

This study effectively demonstrated the relevance and appropriateness of the environmental strategies applied to the design of the Golden Lane Estate in order to achieve timeless modernity and adaptability to the social, environmental, and climatic changes that occurred over the past 60 years. The concept of the 'city within the city' and the very careful design of outdoor spaces as well as of the massing, layout, orientation, and material

finishes were analyzed by the students through a combination of fieldwork measurements, observations, occupants' interviews, and digital performance analysis.

This helped to prove the positive environmental attributes of the estate which seamlessly transitioned from outdoor to indoor spaces and which responded very well to the demands of an increasingly dense, noisy, polluted, and overheated urban context. This was evident especially in the measurements of sound and air quality, which proved that the introverted nature of the estate, developed around courtyards facing away from the busy street front, were conducive to a quieter, cleaner, and sometimes more pleasant microclimate.

Outdoor solar and wind studies helped to demonstrate that, as buildings are spaced by a minimum of 1.5 times in height, wind is allowed to flow into the courtyard without creating turbulences and overshadowing is minimized, allowing good solar access to most blocks. This was shown to contribute to the outdoor comfort, which in all the courtyards has been found, during the monitored typical conditions, to produce no thermal stress according to the Universal Thermal Comfort Index (UTCI).

For the indoor spaces, the students studied the thermal performance of the complex comparing typical apartments in various blocks. This was carried out with a combination of thermal monitoring and simulation. The significant difference in the heating energy demand between two chosen apartments in the Bayer House and the Basterfield House (Figure 3) was found to be due to the retrofitted insulation of the building envelope of the latter with an overall improvement in a weighted average U-value by 17% [44].

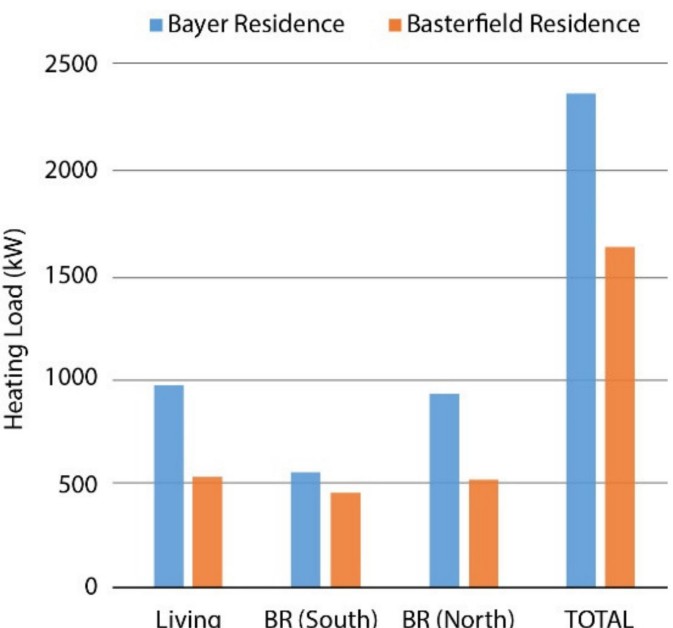

**Figure 3.** Heating loads comparison between the Bayer House and the Basterfield House (source: Deependra Pourel, student 2016–17).

The architects designed the building to best suit the climate and followed the basic principles of having a high glazed area on the south to receive solar gain and with minimal glazing and thick walls on the north to reduce heat loss during the cold winters. Nevertheless, thermal insulation in walls and glazing was not part of the technological advances available or understood in the Modernist period. Hence, as proven by the simulation analysis, the determining factor for achieving energy efficiency and minimizing space heating demand was in fact the retrofit of insulation. The drive to maximize daylight and natural ventilation is apparent in the small details of window design and maximization of glazed and opening areas. However, these present substantial heat losses and high infiltration due to deterioration and aging; however, with the advances of contemporary materials, technology and specification could be improved while retaining very appropriate and forward-looking strategies for climatically responsive buildings.

The study included a detailed solar irradiation analysis [45] performed using computer simulation software [47], which highlighted the role of overhangs on the south façade relative to the various seasons' sun angle and demonstrated the effectiveness of the chosen 0.29 m overhanging projection.

Additionally, environmental monitoring, to collect a dry bulb temperature, relative humidity, and air velocity, as well as illuminance and sound, was performed using point-in-time meters (Figure 4) and Tiny Talk dataloggers [48]. The monitoring highlighted the temperature fluctuation and distribution around the house, with the north bedroom being the room with the highest fluctuation in winter due to intermittent heating and conspicuous heat losses. Temperature stratification was also observed in the south-facing living room and stairs due to a combination of solar gains and grills in the first-floor master bedroom floor, allowing heat from the ground floor radiators migrating upwards. The fieldwork also evidenced the occupants' use of adaptive methods, such as increasing their clothing insulation (measured through the clo-value) to keep themselves comfortable with very few hours of auxiliary active heating.

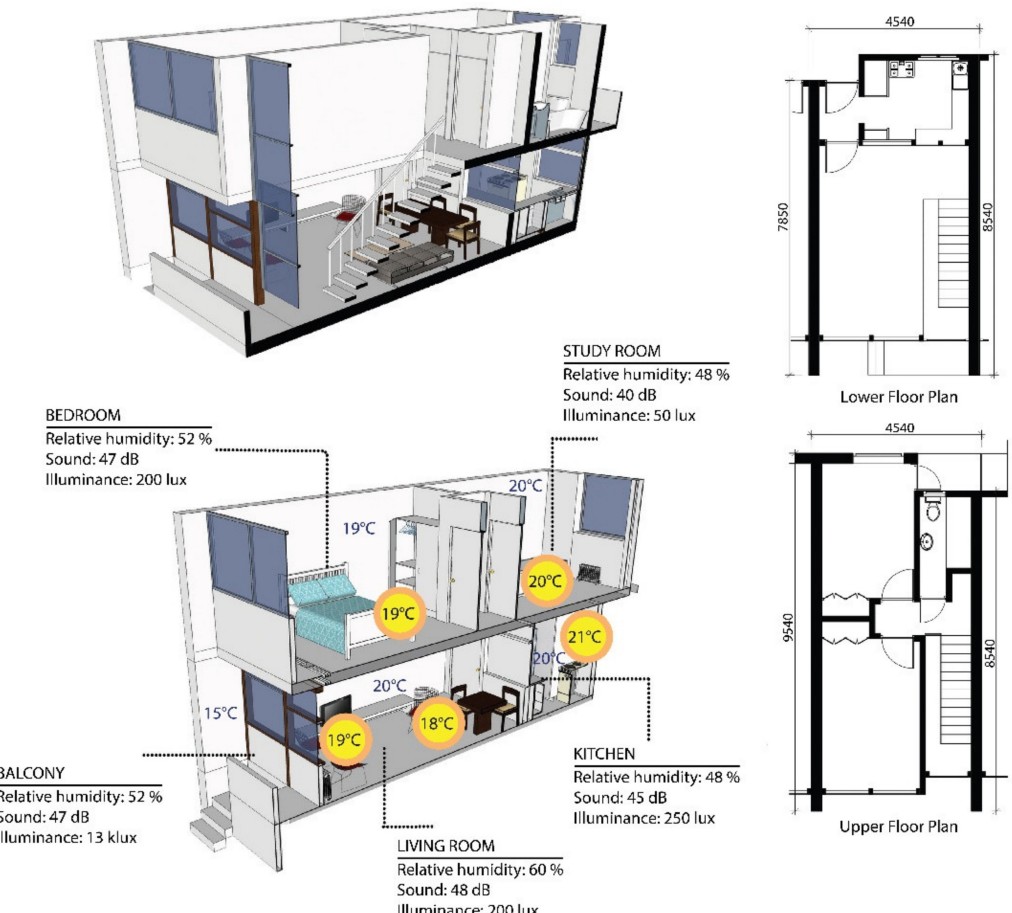

**Figure 4.** Plan and axonometric view of the Bayer House's typical maisonette with point-in-time environmental measurements (source: Deependra Pourel, student 2016–17).

The lessons learned from this case study entailed the positive environmental effect coming from sound strategic design, such as the decision to create an introverted urban space where the community can find respite from the busy, noisy, and nowadays polluted outer streets. Furthermore, a large proportion of the outdoor area, devoted to green and recreational spaces, has tangible benefits in reducing the urban heat island effect, and at the same time creating sufficient exposure to solar radiation in winter. The low–medium-rise building slabs with a north–south orientation for the family houses and east–west orientation for the smaller apartments respond well to the requirements of the climate

and of the occupants' lifestyle and occupancy patterns. The larger glazing areas on the south-facing aspects, coupled with the exposed precast concrete stairs and large opening areas, enhance the opportunity for both passive solar heating in winter and convective cooling in summer. These are some of the lessons learned from the semester one case studies from London, which are then applied as design principles to the semester two Design Projects.

### 5.2. Design Project

Following the Evaluation Studies, a Design Project is developed for a specific site, not necessarily in the same location, dealing with the challenges and potentials of the cultural, climatic, and physical context, including those imposed by the future of climate change. The aim of the design proposals is to respond to the environmental and spatial requirements of a specific program of activities, whilst creating a bespoke/specific architectural response. Often, the sites chosen for this design exercise are in urban areas which also requires consideration of socio-economic issues and other environmental challenges beyond the climatic conditions, including urban heat island effect, as well as air and noise pollution.

Although the practice of refurbishing existing buildings is addressed in the AED master course (the case study of this article), and is discussed and presented to the students as a most definitely sustainable approach, the pedagogic objective of the Design Project module is that the students develop a design proposal from the outset, in other words, "from zero". This is because the students will have to face all the usual design challenges in a totally new project that can be informed by environmental principles, starting from the site planning of the new design proposal and the intrinsic mutual environmental impacts between new building and existing surroundings, which could be determinant factors for the final proposal of form and orientation, façade design, and internal distribution of spaces. In this way, the students can make maximum use of the design application of the environmental design principles (and analytical tools) which were taught in semester one. In addition, although the design proposal has not been one of retrofit or refurbishment, in the recent years (except for the first pandemic year), the site of the building studied in the previous module in semester one (Evaluation Studies) was chosen as the location for the design exercise, and in semester two following a similar program of activities and building types. This is so that the students can benefit from the knowledge generated in semester one on the site conditions, but have the freedom to propose the best possible environmental design solutions, learning from the problems and qualities found in the existing case study, but without the constrains of any existing condition, e.g., building form or orientation. However, students work from previous years have shown that even working in different sites, a great deal of lessons learnt from the semester one Evaluation Studies can be easily and clearly applied in the semester two Design Project module, as long as the program of activities is kept the same. As such, design lessons are related to the environmental response of buildings and sites within a climatic and microclimatic context, and are not only specifically related to their physical site conditions.

### 5.2.1. Didactic Aims and Methods

Whilst the Evaluation Studies mostly required the use of assessment procedures, the Design Project is an exercise that places emphasis on proposition, following a deep understanding of the local context, as well as the specifics and requirements of the program of activities, with the goal to achieve architectural expression of environmentally adequate architectural solutions. In sum, the design exercise combines technical and analytical know-how with creative thinking and a context-sensitive design approach. This concerns a design process that explores future visions for the built environment, working with the possibilities inherited in lifestyles and the socio-dynamics of the place, as well as the impact of form, spatiality, and material on the creation of livable, healthy, and inviting spaces, in specific local environmental conditions, as also proposed in other environmental design studies, based on the evidence-based approach [49].

Methodologically, the purpose of this design exercise is the combination of three learning/practical activities: the application of lessons learned from precedents (including examples of previously developed evaluation projects), the application of environmental principles in the formulation of preliminary design concepts (theoretical knowledge), and the use of analytical procedures with the support of advanced computer simulations to perform feasibility studies and design optimization methods. In this context, as explored in methodological proposals found in the theoretical background [41–43], the use of analytical tools plays a key role in the early design stages, assisting in the formulation of the design concept by testing the validity of initial ideas with parametric and sensitive analysis, and later verifying and improving the performance of the final design proposal [42]. Whenever possible, fieldwork activities, including on-site measurements, interviews, and observations, can add a great deal of information on the local conditions.

As an outcome, the design proposal is meant to respond to spatial quality and environmental performance targets and recommendations, relating the architectural solutions with the local climate and microclimate, user expectations, and specific functional requirements, by considering the impact of the surrounding context upon the design intervention and vice versa. Therefore, for urban sites, the design proposal encompasses the surrounding urban realm and its physical, environmental, and socio-economic conditions. More specifically, the design process explores internal and external spaces in detail to test out the environmental performance of the proposal.

The design brief is meant to be further detailed by the students during the initial stages of the project in parallel to the site analysis and the understanding of the socioeconomic and urban context, alongside the development of preliminary analytic studies of the initial design concepts. It is expected that the environmental opportunities and constraints of the local context not only inform the design solutions, but also key aspects of the brief, including the type of activities to be considered and the definition of areas per activity. Current and future trends associated with the social, environmental, and spatial requirements brought by the pandemic, as well as climate change projections, are also meant to inspire the design brief [37,38].

Once the brief is defined, the very first step of the design process is the elaboration of a matrix with the proposed indoor and outdoor activities and their respective environmental requirements, accompanied by a detailed fieldwork, including measurements of environmental variables on-site (including air and globe temperatures, humidity, air velocity, illuminance levels, and noise levels), if fieldwork is possible. In addition, the site analysis includes analytical assessments of solar access, sky view factors, and air flow within a specific site and immediate surroundings. The findings from these initial studies, coupled with the environmental performance targets of the brief, provide the starting point for the formulation of design research agendas, leading to preliminary design proposals.

The design development is based on a parallel process between the site planning and overall building form and the design of internal spaces. When considering occupants' lifestyle and occupation patterns, adaptive strategies are to be included, considering the flexibility of building components, such as internal and external fittings and finishing (shutters, blinds, shading, etc.), as well as occupant clothing and adaptive behavior [50]. The project is used to demonstrate required spatial and elemental adaptability, flexibility, and required change when designing the building elements and components, from which thermal and solar optical properties are to vary according to the climatic conditions and occupant requirements. Several types of analysis are undertaken during design development to progressively inform the evolution of the design concept and strategies.

At early design stages, rules of thumb [51], manual and steady-state calculations (e.g., daylight, natural ventilation, passive zone depth, heat loss calculations, etc.), are applied. They provide quick, valuable, and reliable initial design inputs without the need of heavy computational studies. The results can inform you of suitable window-to-floor ratios, materials, and orientations. At more advanced design stages, tools can provide detailed hourly, daily, or seasonal performance and visualizations of energy flows over two- or three-

dimensional domains that help to refine the design. Dynamic thermal simulation tools allow many output variables related to thermal comfort and building performance to be interrogated, while dynamic daylight calculations provide detailed information regarding the annual amount of daylight available in the building, considering both occupancy profiles and dynamic shading.

Parametric studies drive the optimization process of the specific and determinant environmental design solutions. In this sense, parametric runs are performed to answer specific design questions and understand the relative importance of typical building design parameters. Typically, the building parameters to be varied in dynamic thermal simulations include building construction, environmental strategies, and occupancy dependent variables (e.g., occupancy schedules, internal heat gains, comfort temperatures, etc.).

Outcomes of the analytic work include building monthly and annual energy demand (from heating and cooling in KWh/m$^2$. year), annual daylight availability (using daylight autonomy and useful daylight index metrics), and hourly zone temperatures for selected typical periods (1 week of 1 day) on free running mode to evaluate indoor thermal comfort and overheating risk. Additional detailed information on the building performance (i.e., solar gains, natural ventilation flow rates, conductive heat losses, glare risk, etc.) may also be registered according to the relevance of each variable, within the context of the local microclimatic conditions and functional requirements.

The Design Study is structured to be developed in groups along a 12-week program, in which the first 6 weeks are dedicated to studying climate and context as well as brief and design concepts, followed by an additional 6 weeks devoted to design development and analytic work.

### 5.2.2. Students Outputs

In the academic year of 2018–19, the Design Project of semester two was developed within the context of the international collaboration project entitled *Latitudes*, coordinated by the University of Westminster and that brought together students from the AED Master Course and the Faculty of Architecture and Urbanism of the University of Sao Paulo (FAUUSP), in a dual-city design studio entitled Latitudes Global Studio (LGS), which ran for two consecutive years (2016–2018), with the theme of the Environmental Performance of Brazilian and British Modernist Architecture [52]. This translated into a common brief where lessons learnt from fieldwork on iconic modernist buildings in São Paulo and London were transferred to the evidence-based environmental architectural design studios in both schools. Brazilian students investigated Brazilian buildings and informed results to the UK partners and vice versa, as UK students were assigned to design for sites in Sao Paulo and the Brazilian students for London. The design exercise was supported by field trips to São Paulo and London, characterizing intense periods of academic exchange, including building visits and design workshops.

The identification of further opportunities for environmental improvement, based on performative shortfalls and advancement in technology, including the benefits of exposed thermal mass coupled with shading cross-ventilation during daytime and nighttime ventilation during the warmer period, alongside the need for passive solar heating during the cooler period (published later on by Goncalves [53] and Uzum [54]), informed the design agenda. Focusing on the experience of the UK students, the contemporary re-interpretation of the Tropical Bioclimatic Modernism in the São Paulo showed considerable potential for improving comfort conditions, energy demand reduction, and other environmental benefits (including air quality). A successful example of that is the design project entitled *The Corner House*. The project's objective was to deliver an evidence-based architectural proposal which would not only respond to the environmental challenges of climate and site, including air quality, but also tackle wider urban issues.

Scientific knowledge on the climatic conditions of the humid subtropical climate of Sao Paulo (Cfa) [55] and the direct experience of the site microclimate showed the importance of air movement and fresh air in achieving comfort by passive means. On

a 30 °C DBT afternoon site visit, comfortable outdoor conditions were experienced by the students under the street shades with 1 m/s wind (sea breeze). These observations inspired the formulation of a design proposal for a "super-environmental" urban housing typology, which aimed to adopt the ecology of the traditional Sao Paulo suburban housing in the context of tall buildings for typical local urban morphology. The concept aimed to create a building microclimate which can mediate the transition between open and indoor space/environment, from the noisy and polluted urban context of Sao Paulo to a healthier and more comfortable indoor environment. Such a microclimate is produced by, and manifested as, common outdoor spaces and private gardens that homeowners as well as users of commercial spaces in the building can enjoy. This led to the architectural concept that combined the spatial and environmental qualities of low-rise residential homes based on the corner house typology in Sao Paulo, with high-level vegetated terraces to denser and taller buildings. Vertical sky components (VSCs), irradiation, and shadow simulations were used to identify opportunities to chamfer and carve the extruded plot, gradually giving rise to the final form of the tall residential building (Figure 5).

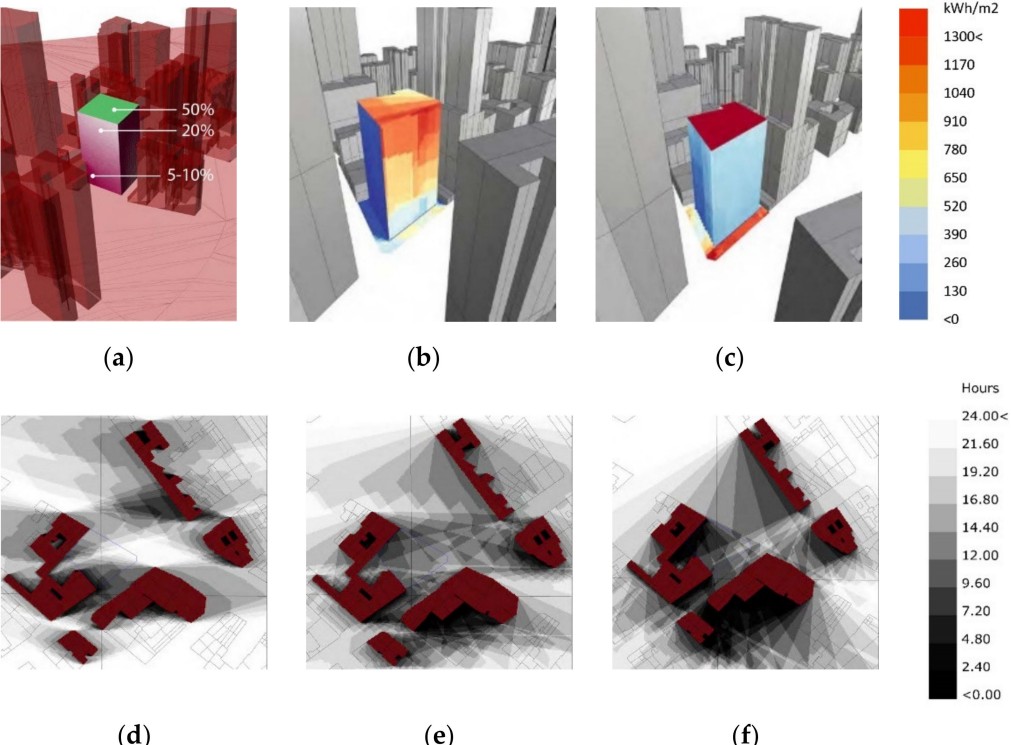

**Figure 5.** Site analysis: (**a**) vertical sky component; cumulative solar irradiation in (**b**) winter and (**c**) summer; sun hours in (**d**) summer solstice, (**e**) equinox, and (**f**) winter solstice. Simulation studies were carried out with the software Grasshopper and the plug-in Ladybug (source: Phung Hieu Minh Van, student 2018–2019).

The main criteria for the chamfering and carving exercise were based on daylight, as well as solar access/control in all apartments. The aim was to find a balance between blocking excessive solar gains (especially on the higher floors) and allowing access to daylight (particular on the lower floors and deeper into the site), in order to fulfil requirements of occupants and vegetation (Figure 6). Secondary to these considerations, but still architecturally and environmentally important, was the provision of good views towards the city. These considerations led to separate blocks linked by shared large gardens every three floors and a massing which could allow the morning sun to penetrate deep into the core of the building.

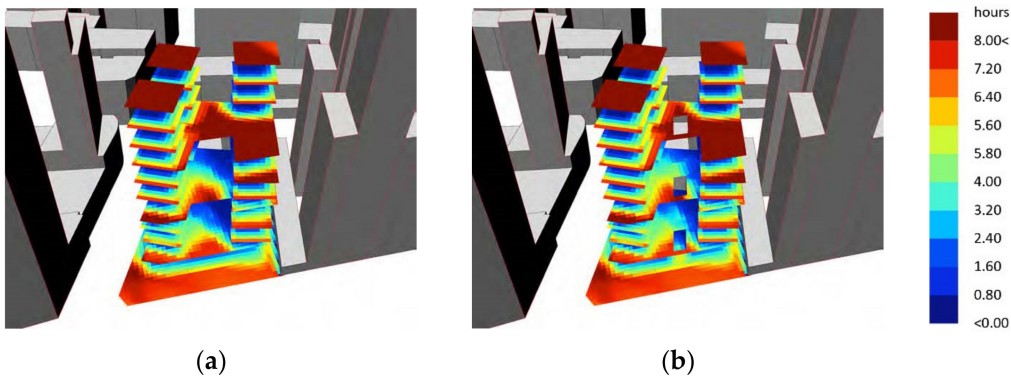

(**a**)　　　　　　　　　　　　　　　　(**b**)

**Figure 6.** Annual sunlight hours for each floor: (**a**) design without lightwells; (**b**) design with lightwells to increase sunlight penetration. Simulation studies were carried out with the software Grasshopper and the plug-in Ladybug. Author (student): Phung Hieu Minh Van.

On a more technical level, the use of vegetation to purify the air at the immediate microclimatic scale was justified and detailed with the support of scientific data [56] that proved that a single plant pot of specific species can purify a volume of 36 m$^3$ and that roots and soil of plants have the potential to purify up to eight times more than leaves. Based on that, the landscape design included vegetated terraces around the living areas and potted balconies with the so-called air-filtering plants in every window (Figure 7).

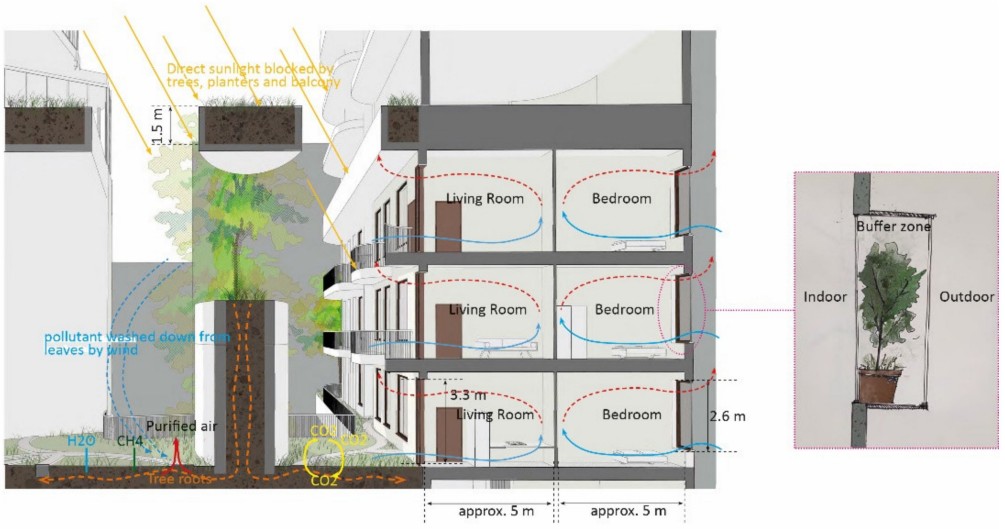

**Figure 7.** Strategic insertion of vegetation to modify microclimatic conditions and improve air quality (source: Phung Hieu Minh Van, student 2018–2019).

### 5.3. Thesis Project

As the final and most summative of the projects in the course, the Thesis Project brings together research methods from the Evaluation Studies and the Design Project, therefore encompassing techniques from the so-called live project, evidence-based design approach, and integrated studio, as described in the literature review [5–8]. The thesis topics vary from themes associated with the environmental performance of the built environment (including outdoor spaces and buildings' design conditions), to the exploration of methodological design approaches.

#### 5.3.1. Didactic Aims and Methods

The projects related to issues of the environmental performance of the built environment configure the Design Study, to which the outcomes refer to design applicability, and Design Project, to which the outcomes refer to design application. The design applicability

is associated with the formulation of design recommendations and guidelines, presented as design strategies and simplified architectural solutions, applicable to the conditions addressed in the initial research questions. Differently, the design application is focused on a pre-selected design situation, resulting in a specific design proposal. Despite the different types of design response (being the first one of a generic nature and the second one site-specific), both research paths are informed by a clear evidence-based approach. In both cases, the formulation of research questions and hypothesis are often climate-specific and driven by the ultimate objective of exploring and demonstrating the environmental attributes of architecture and urban design.

All projects are strongly based on the potential of generalization and examination of hypothetical design scenarios, associated with the possibilities of analytical tools. Fieldwork, or, alternatively, reference to data from existing examples, is particularly important in those projects with emphasis on case studies. Design scenarios and solutions to be analytically studied are drawn from three complementary knowledge sources: (1) principles of environmental design (including lessons learnt from climatic diagnosis and general knowledge of building physics); (2) precedents and fieldwork; and (3) insights from the local context. The Thesis Project is an individual piece of work to be developed over a 12-week period, with an additional 5 weeks for self-guided study and dissertation writing. The students' preferred thesis topics are formulated in the 12 weeks prior to the start of the Thesis Project module.

Each student follows a research agenda featuring a literature review to convincingly contextualize the research topic and precedents; characterize the problems; formulate research questions and hypotheses; collect fieldwork (if applicable) for relevant environmental and/or performance data under predefined environments to evidence the environmental credentials of the outcome; critically evaluate relevant performance analysis and modelling techniques for the applicability of the outcome on a wider range of contexts and scenarios; describe and exemplify the cumulative results; demonstrate the integration of the proposed outcome in innovative workflows for future working environments; present a final conclusion to demonstrate the specialized knowledge gained during the process; and list the research findings, recommendations, and further research objectives.

Through this process, students have the opportunity to explore a topic of interest in detail, adapting and integrating the resources used in Phase 1 and 2 to a narrowed and elaborated research topic. The cultural and geographic diversity of the students' origins is reflected in the dissertation topics, alongside various socio-economic challenges. In this context, the students are asked to embrace and develop an agenda beyond the classic themes of environmental design (daylight, thermal, energy, and acoustic subjects), creating transdisciplinary research and design projects, in order to respond to current and urgent urban and social issues, where environmental design proposals can have a meaningful and positive impact.

Methodologically, in the end, the Thesis Projects are expected to demonstrate advanced applications of analytical tools through the adaptation and optimization of software interfaces and/or workflows, advanced consideration of building physics in computations, and the use of innovative monitoring tools and data-processing techniques.

5.3.2. Collaborative Research

Following the integrated design phase, students face a final stage of research and development in the learning curve where a knowledge confrontation is overcome to master specific areas of environmental design. The process becomes a vehicle for demonstrating the students' cumulative knowledge, as well as developing observational and critical skills gained throughout the pedagogical process. With the objective of producing a research design-based report, the course proves a productive space for the formulation and testing of ideas, briefs, and experimental practice which is supported by peer review, supervision, and tutorials from in-house trainers and, more innovatively, from industry professionals. It is the contribution from this selected expertise that strengthens the link between academic

and professional practice through a close collaboration that can output a piece of work of high relevance to industry by not only responding to cultural, socio-economic, political, or theoretical demands, but also addressing everyday challenges of leading architectural and engineering firms.

This collaborative research model has been running at the MSc AED course for seven years at the time of writing this article, resulting in a wide range of outputs from architectural design optimization to software and tool development. The benefits of this collaboration have been evident from the feedback provided from both 'student-apprentice' and 'mentor'. Students can benefit from taking part in a professional team, i.e., experiencing internal workflows and developing innovative projects with a larger amount of resources, while the industry partners can benefit from the academic expertise and theoretical knowledge which could lead to new methodological approaches eventually applied in their everyday workflows. The academic institution benefits, at the same time, from an increase in diversity of thesis topics, ensuring innovative outcomes that are both publishable and applicable in real practice.

The collaborative research phase starts by defining the topics of common interest for both professionals and academics. It is usually the industry who proposes an area for research and development, in some cases initiated by an in-house research department, and informed by data from real projects, client demands, building regulations, etc. The objectives and expected research outcome within a comprehensive and coherent report can complement (if applicable) a new 'resource' (in the form of a physical or digital tool, interface or workflow), demonstrating the use of appropriate research methods, including fieldwork, performance analysis, and modelling techniques to generate convincing evidence in support of the environmental credentials and energy/carbon reductions in the proposals. The theses developed under the Collaborative Thesis Project program follow essentially the same methodological approach as the non-CTP thesis, with the only difference being increased interaction and exchange with the industry mentors.

### 5.3.3. Students Outputs

In the academic year of 2019–19, two thesis projects went beyond the realm of the more traditional environment and energy topics related to the built environment (including thermal, lighting, acoustics, and energy matters) to look at ways in which nature in the form of greenery can improve urban life. The one entitled *Urban and Building Integrated Vegetation* [56] addresses at the positive impact of vegetation on air quality, with reference to London's urban environment (a field of study originally from the urban-climatology and ecology). In addition, the *Indoor Farming in Future Living Models* project [57], developed in collaboration with Skidmore Owings and Merrill (SOM—London office), explores the feasibility of using a network of hybrid urban farms to deal with the future of food shortages in cities and the opportunities for different scales of urban farming inside and outside buildings, taking the city of London as a base case. Both projects are based on a multi-disciplinary literature review, followed by fieldwork and mainly analytical work, leading to viable design guidelines.

In the first project, the primary aim was to quantify the impacts of vegetation in heavily built areas lacking green elements, and to analyze various typologies of vegetation and their integration within the built environment. One of the key outcomes of this work was the formulation of the concept of urban and building-integrated vegetation (UBIV). The research showed that different UBIV can have a distinct impact on the local urban scale and that the addition of vegetation enhanced thermal comfort, but worsened air quality in most of the existing scenarios due to a reduction in air movement and, therefore, pollution dissipation (Figure 8). One way to mitigate such a negative impact, according to the previous studies and findings presented in this research, is the use of species with lower levels of leaf area density (LAD), thus increasing wind velocity and consequently air quality. Even though the study highlighted that reducing the LAD may not cause a significant change on outdoor comfort, further research should be developed to accurately

study the potential of diverse types of vegetation, in relation to the various configurations of open urban spaces.

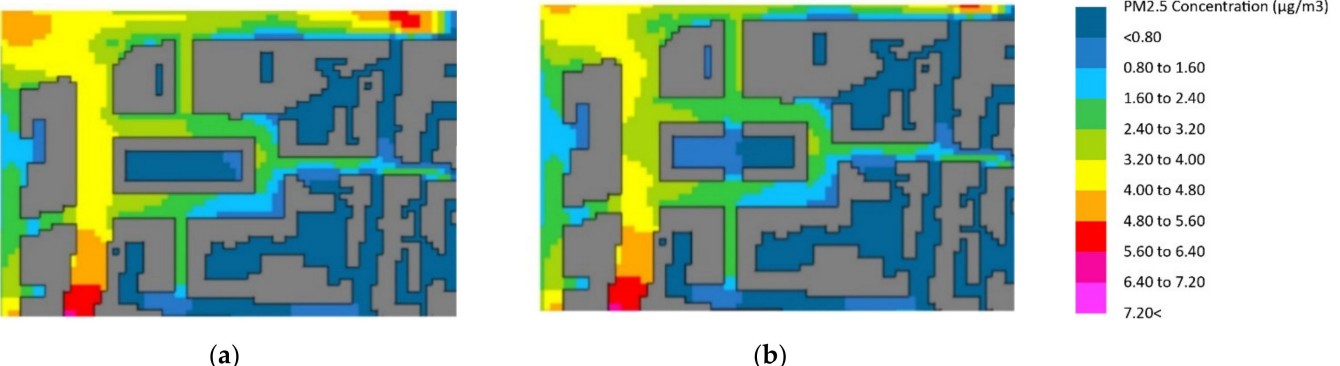

(**a**)                                                            (**b**)

**Figure 8.** Diagram with PM2.5 concentrations: (**a**) scenario of the courtyard building and no vegetation; (**b**) scenario of the U building with living walls and grass. Results from simulations studies carried out with the software ENVImet (source: Joao Matos da Silva, student 2018–2019).

In the second project, the idea of producing food within an urban environment suggests a different new urban lifestyle in the big cities, with challenges regarding the insertion of food production in dense built environments and the achievement of related adequate environmental conditions. In this case, the supporting evidence-based research included daylight and solar analytical studies of three building geometries to evaluate daylight availability and sun hours on the pre-determined potential farming zones within the building (Figure 9). The analysis revealed that the square offers geometry in the optimum exposure.

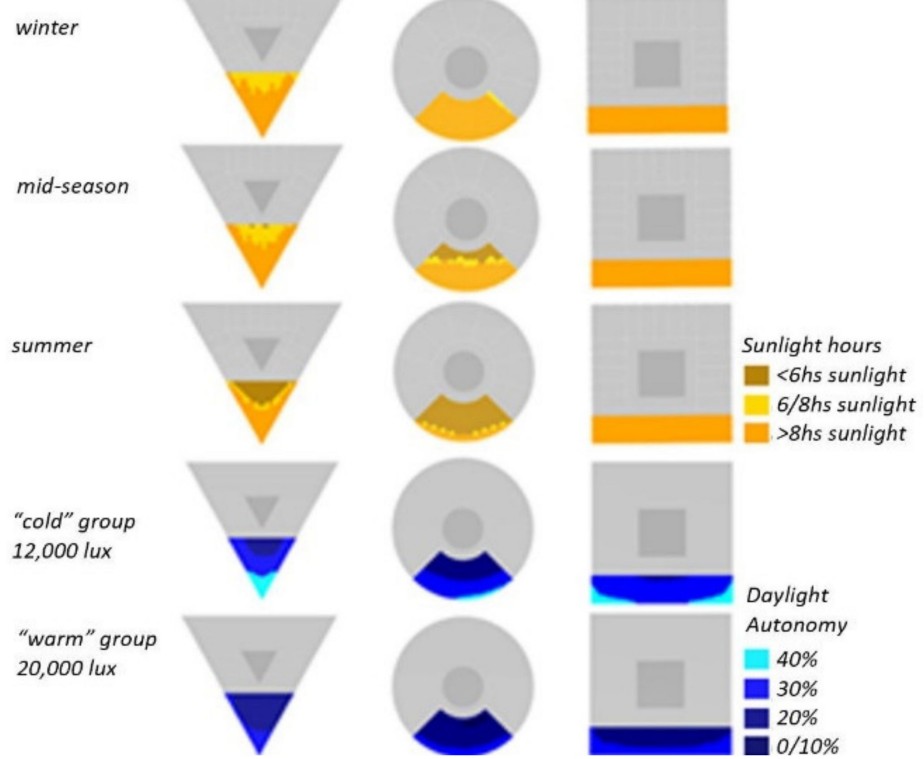

**Figure 9.** Analysis of sunlight hours and daylight autonomy facing the south orientation in winter solstice, midseason, and summer solstice, as well as daylight autonomy facing the south orientation in the 12,000 lux and 20,000 lux thresholds. Simulation studies were carried out with the software Grasshopper and the plug-in Ladybug (source: Carine Berger Woiezechoski, student 2018–2019).

In a second moment of the design explorations, the analytical work aimed to improve the square building form's performance and its final design. The workflow was divided into four main steps, starting by defining the square form's original shape as the base case scenario with a building area of 14,400 m$^2$, where the farming zone's effective production area accounted for 1800 m$^2$. On the second step, the building façade was slightly tilted (until the 10th floor) to optimize sunlight access to the indoor farming, increasing the productive area in 630 m$^2$. On the third step, the respective floor plans on the first floors were also retreated to maintain the residential/office area, whilst also shaping the floor plan along the northern facade for offices, with an aim to optimize diffuse light. Lastly, in the fourth step, the upper part of the southern façade was optimized to create the indoor farm's environmental requirements (Figure 10).

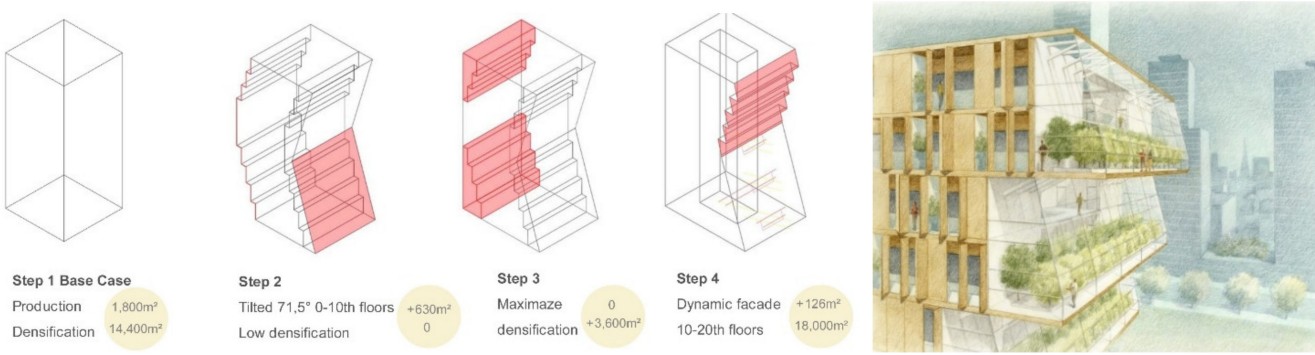

**Figure 10.** On the left—design applicability and form finding process with the views of the north and south orientations for the hybrid farm in the square-based building. On the right—visualization of the design concept (source: Carine Berger Woiezechoski, student 2018–2019).

Moving into the applicability of the analytical results, one vegetable—the lettuce—was chosen for the quantification and comparison of the production potential (alongside other performance criteria, including food miles, energy, and water consumption), among various farming models, i.e., the conventional model, green houses, vertical farming, and the hybrid farming (proposed in this research project in buildings). Ultimately, the design proposal maximizes daylight and sunlight access to produce vegetables using the least amount of energy. In this manner, it was proven that the hybrid farm minimizes greenhouse gas emissions, transportation distances (five times less than the conventional model), and land use.

Given the scientific nature of such research projects, both were developed further in the form of PhD thesis, with the respective following topics: *Small Scale Infrastructure for School Environment,* with interest in the subtropical city of Sao Paulo, and *The Applicability of Hydroponic Farming in Existing Buildings in the Tropical Region.*

The rapid evolution of computational tools for performance simulation over the past five years has offered a recurrent opportunity for the exploration of alternative design methods by means of large data-processing techniques. This has been a part of the research agenda for both academics and practitioners [58,59]. The growing interest in this field from leading engineering firms in the UK led to a collaborative thesis project during the academic year 18–19, introducing an innovative approach to building performance simulation and design decision-making through the development of an environmental information modelling tool (EIM). The work not only aimed to develop a multi-functional tool, but also to solve the drawbacks of unintegrated environmental simulation data while enhancing the communication between design team members in the decision-making process at all design stages, considering a strong necessity in current practice [60–64].

The research aimed to redefine the multi-objective optimization framework that has been recurrent in a number of studies related to building envelope optimization [61] by allowing the adaptable selection of environmental indicators and benchmarks specific to

the project; create an environmental information filter for the customized indicators by applying a weighting system algorithm; and 'translate' the parametric simulation output into a graphic system to easily interpret each environmental solution for the benefit of the design team.

A careful selection of specialized environmental software tools was followed by the mapping of the environmental simulation program that includes benchmarks for each indicator categorized by strategies and architectural features, such as building orientation, massing, zoning, envelope, lighting, ventilation, cooling, and energy. Benchmarks were also categorized by asset and operational rating with reference to most UK standards and certification schemes, including BREEAM, CIBSE guidelines and technical manuals (TMs), WELL building standards, CIBSE TM46, UK energy consumption guides (ECON19), and the real estate environmental benchmark (REEB).

By acknowledging the role of the environmental consultant as the leader of the EIM system, who provides information on the concern of each environmental parameter based on the project brief, the customized weighting system applied to the above-mentioned benchmark results in a score ranging from 0 to 100 that sets the basis of the multi-criteria optimization. A comparison between two different building types (residential and office) demonstrates the composition of environmental indicators, the selection of analysis tasks, and their percentage in the scale for each of them (Figure 11).

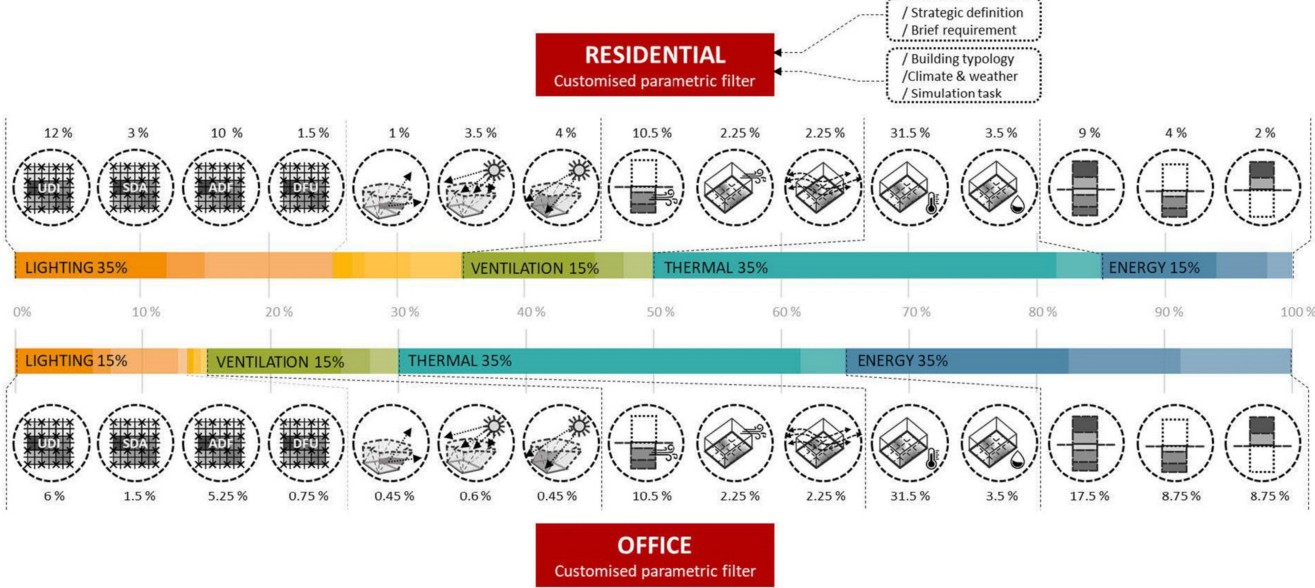

**Figure 11.** Adaptability of customized parametric filter for two different building types. Simulation studies were carried out with the software Grasshopper and the plug-in Ladybug, using EnergyPlus and Radiance as simulation engines (source: Chin Huei Wu, student 2018–19).

The parametric visualization stage aims to provide the overall assessment results by including the final score for each case study and the final scale of the customized impact weighting system, as well as illustrating the environmental impact priority that will ensure the potential improvement space of each parameter (Figure 12).

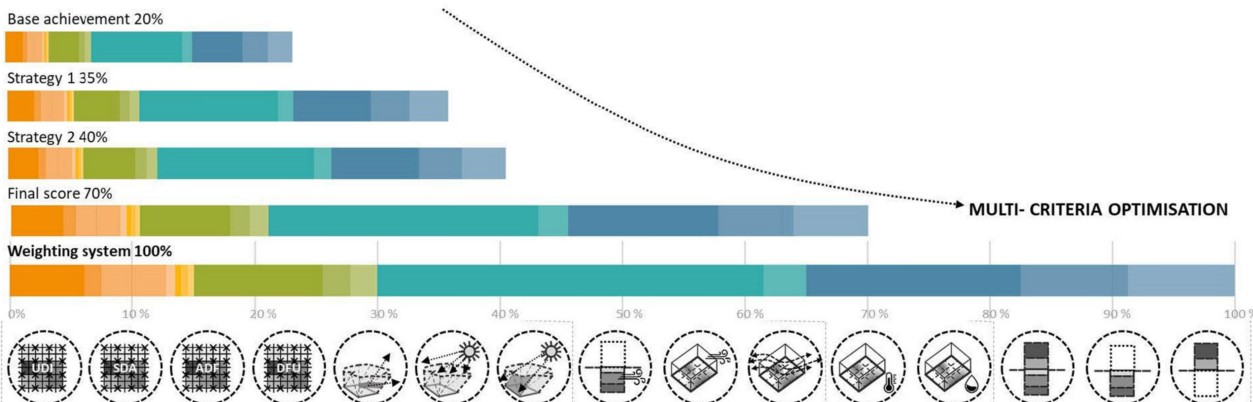

**Figure 12.** Simplified review of EIM tool outcomes. Simulation studies were carried out with the software Grasshopper and the plug-in Ladybug, using EnergyPlus and Radiance as simulation engines (source: Chin Huei Wu, student 2018–19).

The final stage of the research demonstrated the application of the EIM tool for the early design process of an office space. This involved the definition of the parametric impact priority, the strategic effectiveness of each design factor, and the strategic impact priority to allow the final review for the best solution. The illustrated results demonstrate significant differences in building performance and energy consumption between four design options while, at the same time, highlighting the room for potential improvement of each environmental indicator (Figure 13).

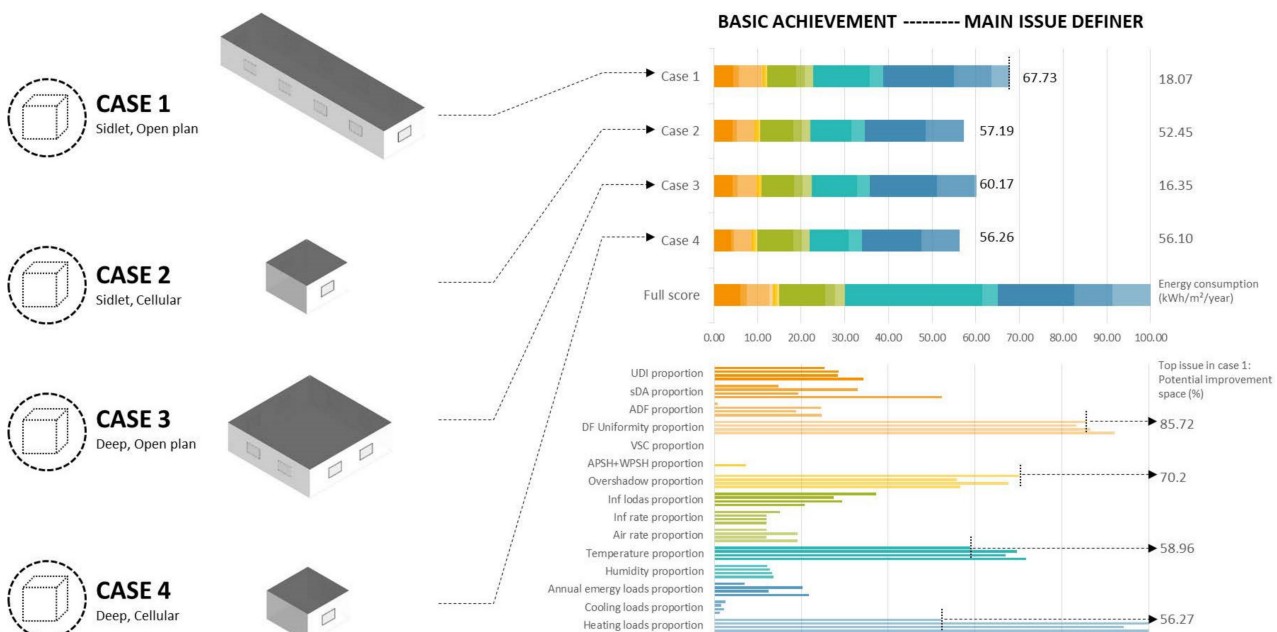

**Figure 13.** Review of the strategic impact priority of four design options. Simulation studies were carried out with the software Grasshopper and the plug-in Ladybug, using EnergyPlus and Radiance as simulation engines (source: Chin Huei Wu, student 2018–19).

The outcome of this research defined a new path for the efficient integration of performative design tools during the design process at all stages and offered new communicative methods. The EIM tool has been used and successfully integrated by the industrial partner Chapman BDSP in several design workflows and the work has been published in international conferences [62].

On the academic year 2019–20, the thesis project, titled *Adaptive and Climate-Responsive Facades for Commercial Buildings in Dubai*, develops a multicriteria parametrical analysis method adapted to the context of Dubai. The thesis responds to the urgent need to address the current energy framework of United Arab Emirates, considering the country with the largest ecological footprint per capita, largely because of its carbon emissions [63]. It also contributes towards research on the decarbonization of the country, i.e., one of the main government objectives by 2050 [64].

The hot arid climatic context of Dubai, characterized by a five-moth warm period with external air temperatures reaching 45 °C and diurnal temperature swings of 20 degrees, together with the evaluation of humidity and pollutions levels, prevailing winds, and frequency of sandstorms, informed a series of bioclimatic strategies to improve operational energy and indoor comfort of open office spaces. Passive evaporative cooling techniques combined with night ventilation strategies and solar control set the basis for the design optimization criteria developed in this research.

The main objective is to inform the design process of adaptive facades that must respond to the current climatic and socio-cultural context by providing a range of recommendations and design solutions with varying energy and daylight outputs. This is achieved by (1) defining a set of parametric variables covering outdoor, indoor, and envelope conditions (such as building orientation, obstructions, noise and air pollution, office layout, massing, internal conditions, material properties of the fixed and adaptable envelope, and shading devices); (2) performing energy and daylight studies of a parametrical model representing an office space for natural ventilation, useful daylight, solar control, thermal comfort, material properties, and noise levels testing; and (3) developing an environmental design matrix that relates design inputs with performance outcomes.

The analytical work was developed in collaboration with the industry partner Hilson Moran, who shares interest in the future design of climate-responsive facades. It consists of the application of a design-oriented multicriteria parametrical input on a computational model representing an office space, including building orientation, room depth, height, number of floor levels, window-to-wall ratios, aperture areas, and shading devices (Figure 14). Performance is interrogated in the form of annual cooling, heating and lighting loads, daylight autonomy (DA), useful daylight illuminance (UDI), spatial daylight autonomy (SDA), indoor wind speed, and annual hours of comfort (Figure 15).

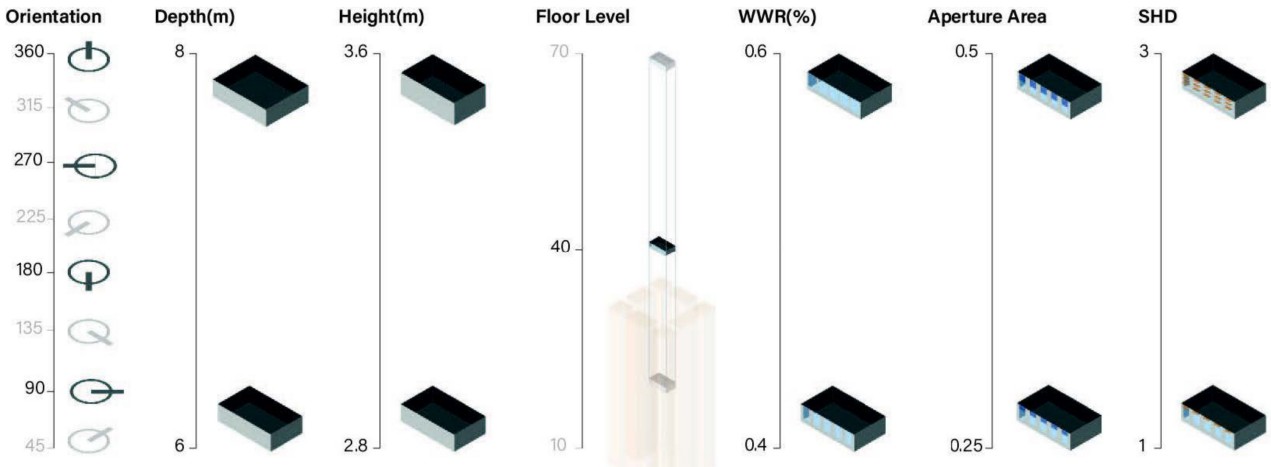

**Figure 14.** The input variables tested in the parametric study. Simulation studies were carried out with the software Design Explorer and Grasshopper, and the plug-in Ladybug, using radiance as a simulation engine (source: Mary-Joe Daccache, student 2019–20).

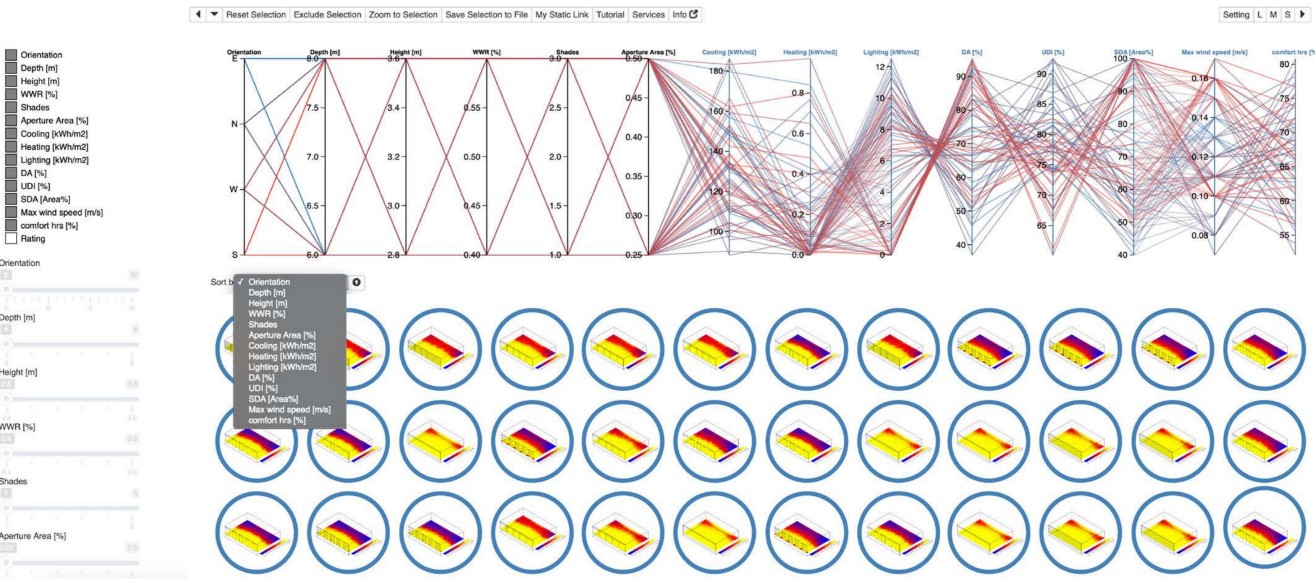

**Figure 15.** Matrix of the design-oriented multicriteria parametrical inputs and outputs. Simulation studies were carried out with the software Design Explorer and Grasshopper, and the plug-in Ladybug, using radiance as the simulation engine (source: Mary-Joe Daccache, student 2019–20).

The integration of emerging specialized environmental software tools within a parametric design engine helped to define a simulation script for the analysis of 128 design iterations that are later processed and visualized using an open-source web interface. With this set of tools, the user can easily interrogate, for instance, the south-oriented design options presenting the lowest range of cooling loads and the highest range of daylight autonomy (Figure 16).

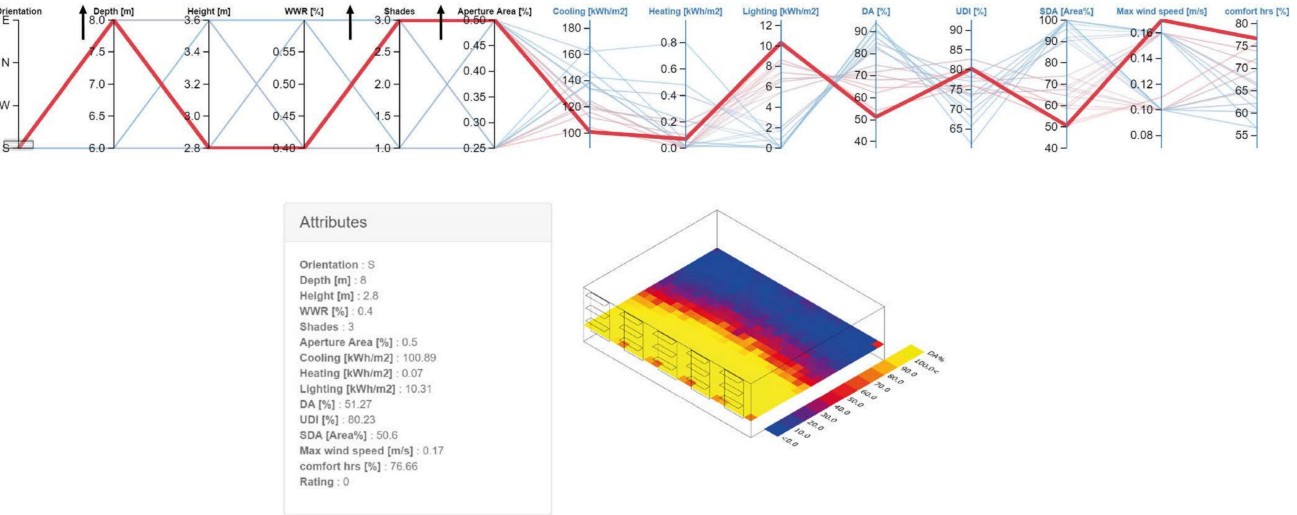

**Figure 16.** Design solution for the south orientation presenting the lowest cooling and heating loads and highest daylight autonomy. Simulation studies were carried out with the software Design Explorer and Grasshopper, and the plug-in Ladybug, using radiance as the simulation engine (source: Mary-Joe Daccache, student 2019–20).

The thesis demonstrated the successful application of the multicriteria analytical method, which has proven to be efficient and valuable. This sets the basis for future research directions that could target a wider range of office typologies as well as design inputs, currently in the research agenda of the industry partner Hilson Moran. This also

reaffirms the continuity and the adaptation of the described research project to future climatic and socio-economical contexts.

## 6. Discussion and Conclusions

Although the examples of student work outlined above are from specific academic years, they were selected as representative of the range of outputs of the students of the course and of the learning outcomes of the main course components as explained in Section 5. Nevertheless, the representativeness and validity of what is presented here is aided by reflections and discussions within the authors and core teaching team, which has remained mostly constant since the course's inception in 2014. Based on these observations and reflections, it is apparent that the gap in knowledge lamented by Marco et al. [6] and Szokolay [7] within the ideal of the integrated studio can be effectively overcome if a strong link can be created between the didactic inputs of the taught modules and the demands of the brief proposed to students in the design studio environment. This link must exist on several levels and factors pertaining to the *what, when, who, and how*. For the *what*, it is easy to assume whatever environmental design content will suffice, but it is fair to say that the teaching staff must resist the temptation to introduce too much content. Even if a very steep learning curve is inevitable (especially when dealing with students with no prior knowledge of the subject), a gradual approach following the intended design or analytic methodology is preferred. For the *when*, the timing of thematic workshops and expected design outputs must be aligned to give a temporal cue to students, creating an expectation that what is introduced in one week needs to be applied in the following. The *who* is an underestimated aspect in many architectural schools so far, but the authors share Szokoloy's [7] strong conviction that the lecturers involved in the teaching of the principles and the analytic/quantitative tools must also be present in the design studio or, ideally, that the numeracy skills of those staff are shared also by the design studio tutors. This is a fundamental requirement for the achievement of a fully integrated approach and avoid the disconnection often present in many design studios. The *how* is probably the most debatable aspect of the discussion as it will vary greatly based on resources, background, didactical choices, and factors which then form the specific milieu and identity of each course. However, one axiom that the authors agree on is that, given that architecture in general is inherently eclectic and interdisciplinary, the true differentiator between a more traditional approach and the approach followed by this course is that of the evidence-based, numerical, and computational quantification as a primary driver for the design. The question that often comes up at this point is whether the evidence-based analytic approach hinders or enhances creativity, and the jury is still out on that. From the perspective of this course, it can be observed that although creativity is somehow downplayed in the early or priming stages of the students' learning experience, this comes back when the analytic tools have been mastered and when a greater confidence is acquired in balancing knowledge and application. This usually happens during the final thesis stage, where a greater level of maturity allows the students to display two types of creativity: the one in which the analytic process shapes the design process (such as in the indoor farming project illustrated above [57]) or the one in which creativity is expressed through problem-solving and dexterity in tackling complex analytics (such as in the thesis on the conception and implementation of an environmental information modelling framework [62]). The downplaying of the creative and design aspects can be seen as a disadvantage for an architectural course which is heavily reliant on numerical computations, also evidenced by outputs focusing more on the robustness of the performance analysis rather than the representation of design features. This is, however, often celebrated and accepted as a necessary trade-off required in a technical course with a limited timescale.

Complementary to the integrated studio and the evidence-based approach is the integration of the live project in environmental design courses and how this can aid and enhance the climate literacy of students. As illustrated by the range of design

studio projects in the course, the live projects are mostly realized through engagement with the occupants of the buildings subject to fieldwork analysis and post-occupancy evaluation in semester one, but also through engagement with the collaborative thesis industry partners, who offer students the opportunity to actively engage with practice and experience professional working methods and environments. It has been observed that, although the industry partnership was initially approached as an opportunity for companies to come in and offer topics of research for students to take on, the actual relationship between academia and industry has been osmotic, with a balanced and reciprocal open-mindedness regarding topics of mutual interest driven by genuine knowledge transfer aims. The same can be said, to a lesser extent, for the relationship with occupants/clients in the semester one projects, when there is open mindedness and welcoming attitude towards the students visiting and analyzing their building, but more tentative outcomes are often achieved due to the initial stages of the students training. Nevertheless, the engagement with real buildings in use is an excellent vehicle for meeting the teaching and learning objectives of the course and for many students to acquire, for the first time, a feel for the environmental and energy parameters which are at the base of the climate literacy learning curve.

The reflections shared above, although specific to the presented pedagogical case study of the AED course, are representative of the challenges and opportunities applicable to many similar postgraduate courses in environmental design and sustainable architecture in the UK, Europe, and beyond. Although the HE's funding mechanisms and management approaches between different countries might differ, the worldwide drive for open access knowledge and information (including open source software, DIY, and portable and mobile app-based sensors) and the wider use of online and remote teaching platforms can provide much greater accessibility to climate literacy content for students. However, it is important that the fundamental nexus between principles of environmental design/building physics and their application in design projects are appropriately guided. This requires academic staff with the right skillset and expertise to be resourced and employed in the design studios. This is a fundamental requirement for students to go beyond the seductive appeal of fancy plots or colorful arrows and, in this way, sadly, promoting green wash, but instead to follow a robust evidence-based numerical and analytical approach where every design step is justified and backed up by properly understood quantitative data and qualitative insight.

With regards to the evidence-based environmental architectural design studio, in addition to the general high-level quality of the students' outcomes, the collaborative thesis project with industry widened the context of student experiences and proved to enhance their level of satisfaction with the AED course (which scored overall levels of satisfaction averaging 82% over 7 years, based on student module evaluation surveys administered by the University of Westminster's KPI Enhancement Office every year [65]). This positive outcome credits the enhanced learning potential of a multilevel pedagogical approach (Figure 17), combining, on an international platform, integrated studios, live projects, and evidence-based methods for improving the teaching, learning, and practice of environmental design in the 21st century. Further research in the field should include quantitative analysis of the impact that the pedagogical approaches described in this article can have on courses both at the undergraduate and the postgraduate level, and the implementation of metrics for the quantification of improved climate literacy before and after scenarios.

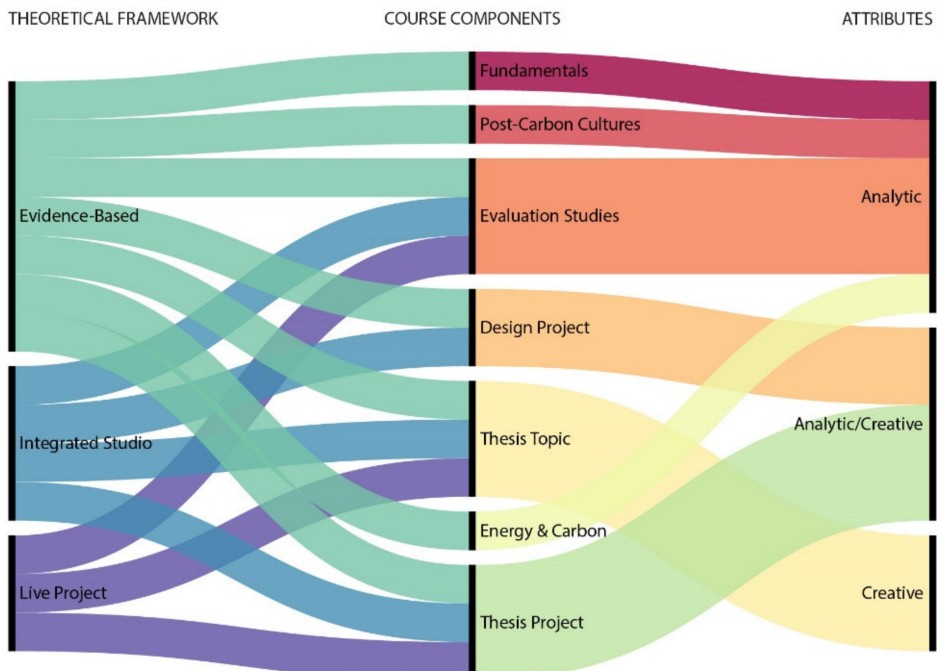

**Figure 17.** Alluvial diagram showing the multilevel pedagogical approach of the course and its components linked to the main reference theoretical frameworks and the spectrum of analytic–creative attributes.

**Author Contributions:** Conceptualization, R.S.-P., J.C.S.G. and J.A.V.; methodology, R.S.-P., J.C.S.G. and J.A.V.; investigation, R.S.-P., J.C.S.G. and J.A.V.; writing—original draft preparation, R.S.-P., J.C.S.G. and J.A.V.; writing—review and editing, R.S.-P., J.C.S.G. and J.A.V. All authors have read and agreed to the published version of the manuscript.

**Funding:** This research received no external funding.

**Institutional Review Board Statement:** Not applicable.

**Informed Consent Statement:** Not applicable.

**Data Availability Statement:** Not applicable.

**Acknowledgments:** The authors would like to acknowledge the students of the MSc Architecture and Environmental Design, School of Architecture and Cities, University of Westminster, who have worked on the projects described in the article: Deependra Pourel, Sharmeen Khan, Minh Van, Carine Berger Woiezechoski, Joao Silva De Matos, Chin Huei Wu, and M.J. Daccache. Thanks go to the course staff and the industrial partners of the Collaborative Thesis Programme for their support to the course and students in developing interesting and relevant thesis projects every year.

**Conflicts of Interest:** The authors declare no conflict of interest.

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
