# Peer review of "Pedagogy Pro-Design and Climate Literacy: Teaching Methods and Research Approaches for Sustainable Architecture"

_sustainability, doi:10.3390/su14116791_

Round 1

Reviewer 1 Report

The article deals with an extremely important topic. The authors are certainly specialists in the field presented. Numerous examples and programs have been cited. The element that is most missing in the text is the overall diagram (or there are diagrams). The description covers many topics, an illustration connecting them and arranging them into stages is NECESSARY! It should be shown that the authors control the whole. The description should follow the logic of the schema.  The structure of the description must be clear, now it is not.

The form of editing the text does not encourage reading. 

Among the disadvantages are the disorderly illustrations. The presentation gives the impression of patchwork. Precision is required in signatures. If the illustration consists of several elements, they must be described (a, b, c). Legends and lettering are scaled differently. The quality of some of the visuals is poor. Some are completely illegible. In the graph in figure 2, please correct the composition of the captions. If we present a decline, e.g. by 30%, we present the same (consistently) everywhere. Figure 1 on page 9 should be Figure 3. Figure 6 is redundant. Figure 8 has an illegible legend. The graduation should enable reading the scale. Please indicate with the initials which student is the author of the illustrations. Some drawings require a signature indicating the software in which they were made. Perhaps you need to introduce a list of abbreviations used? 

Other remarks:
-What AY 2020-21 is (page 5)? 
-(Error! Reference source not found.) How after reading by three Authors that’s possible? 
-Yellow fragment on page 10 
-carelessly edited references: DOI / doi; 9 (1) or 9 issue 1; accessed or accessed on; available or available on or available online etc… 

Due to the lack of structuring of the text and shortcomings in the creation of narratives, I cannot positively assess the innovativeness of the program.

In the discussion, no disadvantages were identified. It is unclear how 90% satisfaction was measured (p.23). 

I congratulate the Authors on their passion. What they do is very important. The manuscript as it stands is unacceptable. Please consider how to better present it.

Author Response

Dear Reviewer, Please see your comments and our responses (in Italics) below.

REVIEWER 1

The article deals with an extremely important topic. The authors are certainly specialists in the field presented. Numerous examples and programs have been cited. The element that is most missing in the text is the overall diagram (or there are diagrams). The description covers many topics, an illustration connecting them and arranging them into stages is NECESSARY!

Response: Thank you for your suggestion. Two diagrams have been added in Figure 1 and Figure 17 respectively.

It should be shown that the authors control the whole. The description should follow the logic of the schema.  The structure of the description must be clear, now it is not.

The form of editing the text does not encourage reading. 

Among the disadvantages are the disorderly illustrations.

The presentation gives the impression of patchwork.

Response: Thank you for your comment and we are sorry your version of the article was scrambled. This formatting issues did not appear in our version; therefore, we will be submitting a PDF along the word version next time. In the meantime, we have improved the formatting generally where we could and following the publisher’s template.

Precision is required in signatures.

Response: Sources of images and authorships have been added.

If the illustration consists of several elements, they must be described (a, b, c). Legends and lettering are scaled differently.

The quality of some of the visuals is poor. Some are completely illegible.

Response: Images quality, elements naming, captioning and legends have been improved or fixed.

In the graph in figure 2, please correct the composition of the captions. If we present a decline, e.g. by 30%, we present the same (consistently) everywhere.

Response: The 30% annotation on the last bar chart has been removed. To be noted is that the cumulative reduction by 30% doesn’t correspond to a reduction by 30% in each of the individual cases.

Figure 1 on page 9 should be Figure 3.

Response: Thanks for warning us of this but the figure numbers are not wrong in our document. Nevertheless, we have rechecked the figure numbers cross referencing.

Figure 6 is redundant.

Response: Figure 6 has been removed.

Figure 8 has an illegible legend. The graduation should enable reading the scale.

Response: A new legend has been added.

Please indicate with the initials which student is the author of the illustrations.

Response: The students names have been added in the source of the illustrations.

Some drawings require a signature indicating the software in which they were made.

Response: The software has been included in the illustrations’ caption.

Perhaps you need to introduce a list of abbreviations used? 

Response: Thank you for this suggestion, however, we don’t think is necessary as all acronyms have now been speeled out. I Spelled out Vertical Sky Component in one figure caption.

Other remarks:
-What AY 2020-21 is (page 5)? 

Response: AY stands for Academic Year. All references to the acronym AY have been spelled out into Academic Year.

- (Error! Reference source not found.) How after reading by three Authors that’s possible? 

Response: Thank you very much for alerting us to this problem and for the fine editing. However, we could not find any such error and will now submit a PDF as well as the word document.

-Yellow fragment on page 10 

Response: Not found.

-carelessly edited references: DOI / doi; 9 (1) or 9 issue 1; accessed or accessed on; available or available on or available online etc… 

Response: References and DOI have been checked and corrected.

Due to the lack of structuring of the text and shortcomings in the creation of narratives, I cannot positively assess the innovativeness of the program.

Response: We are sorry of the perceived shortcomings, and we hope that the revised version of the article would allow a better and more intelligible reading of it.

In the discussion, no disadvantages were identified. It is unclear how 90% satisfaction was measured (p.23). 

Response: Clarification on the source of the 90% satisfaction was added on page 25 last paragraph. Disadvantages were added on page 24 last paragraph.

I congratulate the Authors on their passion. What they do is very important. The manuscript as it stands is unacceptable. Please consider how to better present it.

Response: Thank you for the encouragement and we hope that the revisions made would be sufficient improvement.

Reviewer 2 Report

The article presents an interdisciplinary pedagogical approach to teaching and to learning environmental design principles and practices in architectural education applied in the Architecture and Environmental Design MSc course at the University of Westminster (UK). The subject proposed by the article is pertinent, crucial and extremely useful for reflection on how to deal with sustainability issues in architectural education. By now, architecture education should have incorporated the topic of sustainability into the study plans, educational practices and pedagogical approaches of architectural courses. In my opinion, this article gives a significant contribution to this discussion both in terms of contents and of methodological approach. Congratulations to the authors and to the educational programme.

The manuscript is well-organised, easy to read and gives concise and clear amount of information regarding theoretical background supported by an extensive bibliography (8 self-citations in 61 citations), learning and teaching methodologies and approaches, the structure of the MSc as well as the philosophy and practices. The only element that I would suggest adding, is a scheme or a diagram to explain the sequence of the steps/aims/results. It would help to summarize and to understand the overview of the educational programme.

OVERALL RECOMMENDATION

References from page 6 onwards appear with an error message.

Figure from page 9 is number 3 (instead of 1).

Question: Are the students' interventions in the Design Project and Thesis Project stages always reuse/refurbishment interventions on existing buildings/spaces thus promoting a culture of reuse/refurbishment, itself an ecological and sustainable architectural approach? This is not clear in the article. If so, why is the Design Project stage not developed on the building/locations studied and analysed in the previous phase (Evaluation Studies) taking advantages of the knowledge acquired?

Without wanting to compromise the quality of the article, I felt, however, the need of clarifying the following topic:

  • How does the MSc deal with the relationship between the environmental sustainability and the social and cultural aspects, taking in consideration not only the local climate conditions but promoting a critical thinking on local cultural identities of the places (including aesthetical, historical, material…) and its relation to the social challenges brought by current conditions of society (new notions of family, community needs, demography, work/education, technology, mobility patterns…) with significant typological, programmatic and functional impacts? To summarize, how does the course deal with the comprehensive approach of the concept of sustainability (environmental, social, economic, cultural, governance)?

Author Response

Dear Reviewer, Please find below in Italics our responses to your comments.

REVIWER 2

The article presents an interdisciplinary pedagogical approach to teaching and to learning environmental design principles and practices in architectural education applied in the Architecture and Environmental Design MSc course at the University of Westminster (UK). The subject proposed by the article is pertinent, crucial and extremely useful for reflection on how to deal with sustainability issues in architectural education. By now, architecture education should have incorporated the topic of sustainability into the study plans, educational practices and pedagogical approaches of architectural courses. In my opinion, this article gives a significant contribution to this discussion both in terms of contents and of methodological approach. Congratulations to the authors and to the educational programme.

The manuscript is well-organised, easy to read and gives concise and clear amount of information regarding theoretical background supported by an extensive bibliography (8 self-citations in 61 citations), learning and teaching methodologies and approaches, the structure of the MSc as well as the philosophy and practices. The only element that I would suggest adding, is a scheme or a diagram to explain the sequence of the steps/aims/results. It would help to summarize and to understand the overview of the educational programme.

Response: Thanks for the suggestion. This has been added in Figures 1 and 17.

OVERALL RECOMMENDATION

  • References from page 6 onwards appear with an error message.

Response: This error does not appear in the file that was up-loaded by the authors.

  • Figure from page 9 is number 3 (instead of 1).

Response: This error is not seen in the file of the manuscript up-load by the authors.

  • Question: Are the students' interventions in the Design Project and Thesis Project stages always reuse/refurbishment interventions on existing buildings/spaces thus promoting a culture of reuse/refurbishment, itself an ecological and sustainable architectural approach? This is not clear in the article. If so, why is the Design Project stage not developed on the building/locations studied and analysed in the previous phase (Evaluation Studies) taking advantages of the knowledge acquired.

Response: The answer to this question is “no” in the case of the Design Project and “eventually yes” in the case of the Thesis Project. On the contrary, although the practice or refurbishing existing buildings is addressed in the AED master course (the case-study of this article) is discussed and presented to the students as a most definitely sustainable approach, the pedagogic objective of the Design Project module is that the students develop a design proposal from the outset, in other words, “from zero”. The reason why is because in a totally new project the students will have to face all the usual design challenges that can be informed by environmental principles, starting from the site planning of the new design proposal and the intrinsic mutual environmental impacts between new building and existing surroundings, which could be determinant factors for the final proposal of form and orientation, façade design and internal distribution of spaces. In this way, the students can make maximum use in terms of design application of the principles of environmental design (and analytical tools) which were taught in Semester 1.

In addition, although the design proposal has not been one of retrofit or refurbishment, in the recent years (except for the first pandemic year), the site of the building studied in the previous module, in semester 1 (Evaluation Studies), have been chosen as the location for the design exercise, in semester 2, following a similar programme of activities and building types. This is so that the students can benefit from the knowledge generated in semester 1 about the site conditions but have the freedom to propose the best possible environmental design solutions, learning from the problems and qualities found in the existing case-study, but without the constrains of any existing condition, for example – building form or orientation. Despite the repetition of the site in the semester 2, in recent years, students’ work from previous years have shown that even working in different sites, a great deal of lessons learnt from the Evaluation Studies can be easily and clearly applied in the Design Project module, as long as the programme of activities is kept the same, as such design lessons are related to the environmental response of buildings and sites to their climatic context and not only specifically to their  physical site conditions.

It is worth clarifying that the examples of course work included in the paper for semesters 1 and 2 – meaning Evaluation Studies and Design Project, are not from the same year, for this reason, the two pieces of students’ work are not linked, as it normally happens in the flow of a specific academic year. This was done to give an idea of the variety of projects that have developed in the course, including projects in sites outside London (where the course is located), in different climates – in this case, the city of Sao Paulo, in Brazil. Nevertheless, to show the application of the acknowledge acquired in semester 1 and the link between academic semesters, when presenting the example of Design Project, the influence of lessons learnt from semester 1 were explicitly mentioned in the text.

With regards to the Thesis Project, as this is a choice of the student, the theme of retrofit and refurbishment of existing buildings have been a recurrent one, including a very successful project developed for an office building in London in the academic year of 2019-2020. This one was not chosen to be discussed in this paper, because the intention of the occupants was to present projects that were more transdisciplinary and addressed a wider range of urban, environmental and social issues.

The explanation above has been added in a summarized way in the revised version of the paper, in the presentation of the Evaluation Studies and the Design Project modules.

Without wanting to compromise the quality of the article, I felt, however, the need of clarifying the following topic:

  • How does the MSc deal with the relationship between the environmental sustainability and the social and cultural aspects, taking in consideration not only the local climate conditions but promoting a critical thinking on local cultural identities of the places (including aesthetical, historical, material…) and its relation to the social challenges brought by current conditions of society (new notions of family, community needs, demography, work/education, technology, mobility patterns…) with significant typological, programmatic and functional impacts? To summarize, how does the course deal with the comprehensive approach of the concept of sustainability (environmental, social, economic, cultural, governance)?

Response: The inclusion of a wider design agenda going beyond environmental issues and contemplating cultural, social and economic aspects in general, has been at the core of the key modules of the AED Master Course, being part in one way or another of all modules of the complete academic year.  This is clearly stated in the article, in “The Pedagogical case-study". As a demonstration of that, with reference to the Evaluation Studies, for example, the students are advised to observe and register the “real-life” routines and occupational patterns in buildings and open-spaces selected as case-studies. This encompasses all the means to adapt to certain environmental conditions and the influence of cultural habits and lifestyle on both the perception of comfort and spatial quality as well as on energy demand in buildings. A similar approach is taken to the Design Project, in which the students start by detailing the brief of activities and environmental objectives considering the influence cultural, lifestyle and other real-life parameters.

A clear example of such a methodological approach was the proposal for both modules during the 1st year of the pandemic, when the students analysed in semester 1 the impact and the demands created by the lock-down in the environmental conditions and occupants' comfort and wellbeing in their own homes. Following up, in semester 2 they were then asked to redesign their own residential buildings considering the changes in lifestyle left by the legacy of the pandemic. These particular aspects of the Evaluation Studies and Design Project were inserted in the text of the original submission of this manuscript (3rd and 4th paragraphs of item 5.1.1. Didactic Aims and Methods; 1st and 4th paragraph of item 5.2.1. Didactic Aims and Methods).                  

Moving to the Thesis Project, there are two aspects to be highlighted. The first one is the fact that the course is totally “decolonized”, which means that each student is allowed to investigate, a topic of his/her own interest in any climatic, urban, cultural and social-economic context, from an environmental perspective. Consequently, the cultural and geographic diversity of the students’ origins is reflected in the dissertation topics, alongside various socio-economic challenges, for example, the topic of sustainable social housing is often present among the yearly selection of thesis topics. The second aspect to be mentioned here is the fact that in the Thesis Projects, the students are asked to develop and embrace an agenda beyond the classic themes of environmental design (daylight, thermal, energy and acoustic subjects), creating transdisciplinary research and design projects, in order to respond to current and urgent urban and social issues, where environmental design proposals can have a meaningful and positive impact.

These particular aspects of the Thesis Project were also mentioned in the text of the original submission of this manuscript, and clarifications were added in the revised version (4th paragraph of item 5.3.1. Didactic Aims and Methods).

Reviewer 3 Report

This paper highlighted a variety of pedagogical approaches to the teaching and learning of environmental design principles and practice in architectural education with a focus on recurrent methods applied in specialist curricula in the UK. The topic is interesting and meaningful. The paper is well organized and the written English is fairly good. The conclusions is helpful for the  architectural education in the background of time. I suggested the acceptance of this paper after following my two comments:

  1. A thorough literature review should be added to highlight the novelty of this study. And answer the question: 1) why your proposal is important; and 2) what is the original contribution of the work?
  2. It is better to divide the discussion & conclusion into two separate parts, and limitations of this study should be added.

Author Response

Dear Reviewer, Please find below (in Italics) our detailed responses to your comments.

REVIEWER 3

This paper highlighted a variety of pedagogical approaches to the teaching and learning of environmental design principles and practice in architectural education with a focus on recurrent methods applied in specialist curricula in the UK. The topic is interesting and meaningful. The paper is well organized and the written English is fairly good. The conclusions is helpful for the architectural education in the background of time. I suggested the acceptance of this paper after following my two comments:

A thorough literature review should be added to highlight the novelty of this study. And answer the question: 1) why your proposal is important; and 2) what is the original contribution of the work?

Response: The literature review has been included in the introduction referring to a number of studies on environmental design education, notably that of a research project analysing the state of ED education in Europe. In response to the results of this study we are offering a case study based on our experience and expertise in the field and share the formula of a successful approach based on underpinning critical literature. We have also added new references such as 23, 24, 32 and 65.

It is better to divide the discussion & conclusion into two separate parts, and limitations of this study should be added.

Response: Thank you for the suggestion, however, the template recommends to add a conclusion only when the discussion is too long. Given that this is not an article on a quantitative research project, but a reflective paper based on qualitative research, we feel it is more appropriate to combine discussion and conclusion.

Reviewer 4 Report

This manuscript presents a case study on the how climate literacy can be improved through pedagogical design of education programme. Overall, the language used is very good. The arguments are logical. However, there are some critical issues to be addressed/justified by the authors:

1) The literature review is not broad or comprehensive enough. I know there are many similar Australia-based studies on sustainability issue in construction-related HE programme.

2) The manuscript presents a single case study only. It is rather difficult to draw upon this study to give some solid conclusion about the "usefulness" of the pedagogies.

3) Actually, the second comment is related to the limitations of the research which have not been highlighted and discussed in the manuscript. 

4) There should be some measurement of the "climate literacy". A lot of  previous empirical studies work on the measurement of "sustainability literacy" of construction programme studies. I question why didn't the authors conduct a before-and-after experiment to "quantify" the impacts of the pedagogical design of education programme.

Author Response

Dear Reviewer, Please find below responses (in Italics) to your comments.

REVIEWER 4

This manuscript presents a case study on the how climate literacy can be improved through pedagogical design of education programme. Overall, the language used is very good. The arguments are logical. However, there are some critical issues to be addressed/justified by the authors:

The literature review is not broad or comprehensive enough. I know there are many similar Australia-based studies on sustainability issue in construction-related HE programme.

Response: Thank you for your suggestion. The article’s objectives are not to compare the various educational programmes in different countries as this would have required a different structure and methodology. Also, the focus of our investigation is on architecture courses rather than construction courses, which in the UK belong to two different disciplines, hence emphasising the fact that it is difficult to make a comparison between the UK and Australia, unless this was the primary objective of the article. The article instead tries to respond to the aims of the special issue on Sustainable Architectural Education. Nevertheless, some additional references have been made to other courses in the UK (see section 4 page 3).

The manuscript presents a single case study only. It is rather difficult to draw upon this study to give some solid conclusion about the "usefulness" of the pedagogies. Actually, the second comment is related to the limitations of the research which have not been highlighted and discussed in the manuscript. 

Response: The limitations have been added in the methodology section (section 3, page 2). This research is based on a qualitative method and not quantitative one, and the case study method has been duly referenced (Zainal, 2007).

There should be some measurement of the "climate literacy". A lot of previous empirical studies work on the measurement of "sustainability literacy" of construction programme studies. I question why didn't the authors conduct a before-and-after experiment to "quantify" the impacts of the pedagogical design of education programme.

Response: Thanks for your suggestion, however, as mentioned above, this is a qualitative reflection based on the case study method. The course was created as an environmental design course from the start and it was not possible to create before and after scenarios.

Round 2

Reviewer 1 Report

Thank you for the correction and kind replies.

The corrections made made the article more understandable. However, the structure of the text could have been better. This form makes it possible to understand the whole, even though it is difficult. Such a text can assist academics to modify their approach to teaching project classes efficiently. The individual aspects are not unique. A wide range of topics are discussed, which is  beneficial. Among other information, this report should include the following: the number of lecturers (what were their specializations - just architects? ), the number of students, and the number of hours spent in class. Write down the starting skills the students were expected to possess. What hardware and software will be needed to implement your program? What kind of investment activities did the program require, and how long has it been in development? Feedback and student needs were collected and analyzed for a period of time. This will allow other universities to implement similar programs.

Build a clear framework at the beginning. By providing such information, as well as an outline of what is to follow, the reader will be better prepared to receive the message. Therefore, I propose to modify the chapter methodology in this manner. The introduction of a structured narrative will make reading more enjoyable.

Good luck!

Author Response

Dear reviewer, Thanks for your insightful comments. The detailed responses (in Italics) are as follows.

"Among other information, this report should include the following: the number of lecturers (what were their specializations - just architects?), the number of students, and the number of hours spent in class. Write down the starting skills the students were expected to possess. What hardware and software will be needed to implement your program? What kind of investment activities did the program require, and how long has it been in development?"

Response: Thank you for your suggestion. Changes have been added in section 4 (page 4) – The Pedagogical Case study – as this is the section where the features of the course are presented. Some features of the course could not be provided because confidential or simply not available.

"Build a clear framework at the beginning. By providing such information, as well as an outline of what is to follow, the reader will be better prepared to receive the message. Therefore, I propose to modify the chapter methodology in this manner. The introduction of a structured narrative will make reading more enjoyable."

Response: We have made some small additions in the Methodology section on page 2 to signpost the three main phases of the methodology and cross reference them to the structure and narrative of the article.

Reviewer 4 Report

The authors have addressed the reviewers' comments to my satisfaction.  I don't have any further comments on the revised manuscript except that I would suggest the authors to include an agenda of further research on the topic. The before-and-after analysis should be warranted to find the the real effectiveness of the new model.

Author Response

Dear Reviewer, Thanks for your suggestions to which we have responded below (in italics).

"The authors have addressed the reviewers' comments to my satisfaction.  I don't have any further comments on the revised manuscript except that I would suggest the authors to include an agenda of further research on the topic. The before-and-after analysis should be warranted to find the real effectiveness of the new model."

Response: Thanks for the kind remarks. We have added a paragraph on further research in the discussion section on page 26.